# A Neoteric View of *sp*^2^ Amorphous Carbon

**DOI:** 10.3390/nano13101648

**Published:** 2023-05-15

**Authors:** Elena F. Sheka

**Affiliations:** Institute of Physical Researches and Technology, Peoples’ Friendship University of Russia (RUDN University), 117198 Moscow, Russia; sheka@icp.ac.ru

**Keywords:** *sp*^2^ amorphous carbons, molecular short-range order, necklaced graphene molecules, structural and compositional analytics, virtual spectrometry analytics

## Abstract

Presented is a concentrated synopsis of facilities of empirical and virtual analytics that, once applied, have provided a fully new vision of *sp*^2^ amorphous carbons. This study proved that the solids are multilevel structures, started with the first-level basic structural units (BSUs) and accomplished as macroscopic agglomerates of globular structures, consisting, in its turn, of stacked BSUs. BSUs present necklaced graphene molecules, size, and shape of which are governed by the relevant graphene domains while chemical composition in addition to basic carbon is controlled with heteroatoms of the necklaces. This study shows that BSUs and stacks of BSUs determine the short-range order of the solids and are the main subject of the applied analytics. The synopsis consists of two parts related to empirical and virtual analytics. The former is composed of sections related to structural determination, total and atomic chemical content evaluation and elicitation of the covalent bond composition. The second presents new analytic approaches based on the Digital Twins concept and virtual vibrational spectrometry. The synopsis is configured as an atlas composed of generalized pictures accompanied with necessary explanations to be discussed in detail in the extended references.

## 1. A Concise Historical Introduction

The last two decades have seen a profound breakthrough in our understanding of *sp*^2^ amorphous carbon (aC). First, this was due to the epoch-making development of graphenics—a new material science. The new scientific trend revived a huge interest in solid carbon in general, and gave rise to the first, and then subsequently more convincing guesses that the history of graphenics did not begin in 2010 but goes back centuries ago, and that it is precisely aC that shows the way. The second important circumstance, which stimulated a new interest in amorphous carbon, is the highly elevated level of research into the structure and chemical composition of materials. In the rays of these two illuminations, *sp*^2^ aC began to play with completely new facets. The purpose of this review is to introduce readers to a new vision of this old material.

My first personal encounter with *sp*^2^ aC was in 2005. Being engaged in fullerenes at that time, I was puzzled by the search for evidence of the presence of this *sp*^2^ nanocarbon in nature. Having heard about its possible observation in a natural mineral called shungite carbon, huge deposits of which are located within Russian Karelia, I went to sort it out on the spot. Scholars from the Institute of Geology of the Karelian Scientific Center of the Russian Academy of Sciences, who worked on these deposits, told me many interesting things about this mineral, but its connection with fullerenes has not been confirmed. However, fairy tale-like stories about its outstanding properties, its belonging to the family of nanoscale solid carbons, and the high degree of carbonization reaching more than 98%, did not allow for forgetting this amazing substance. Throughout the formation and further development of the radical concept of *sp*^2^ nanocarbons, which then included fullerenes, carbon nanotubes, and first new graphene materials, such as graphene oxide and reduced graphene oxide (rGO), my attention again and again turned to shungite carbon. After a deep analysis in collaboration with N. N. Rozhkova it was suggested that the mineral is of a complex multilevel structure based on molecular-like compositions, which are nanosize graphene domains framed along the perimeter by heteroatoms that provide complete and/or partial termination of dangling valence bonds of the domain edge atoms [1]. This assumption was in line with fundamentals of chemical processes in nanoscale systems presented in a conceptual article by R. Hofmann, Nobel Prize winner in chemistry [2]. Thus, the idea arose that the basic structural unit (BSU) of shungite carbon is a necklaced graphene molecule (NGM). Analyzing the available chemical content data, we assumed that hydrogen and oxygen are the main components of this BSU’s heteroatom necklace.

At the same time, all geophysicists share common opinion that shungite carbon belongs to amorphous solids, i.e., is a member of the aC family. In turn, this body belongs to covalent solids, in which the short-range order [3] is evidently determined by a particular configuration of covalent *sp*^2^ C–C bonds. Chemistry teaches us that covalent bonds are extremely persistent, are characterized with a well-defined local topology, and tend to strongly resist changing their lengths and angles to ensure a stable chemical composition of substances. Therefore, it seems reasonable to assume that, within the limits of the short-range order space, *sp*^2^ C–C bonds retain their standard topology, thereby separating the graphene domain limited in size from the total massif of an extended honeycomb structure. A further variation in the domain arrangement ensures the disorder of the solid. In this sense, shungite carbon is analogous to covalent molecular amorphics, in which the short-range order is determined with the basic standard molecules [4]. Accordingly, it is natural to assume that the NGMs, suggested to explain the structure of shungite carbon, can be considered as BSUs of *sp*^2^ aC of any origin. Moreover, the well-studied multilevel structure of shungite carbon can be taken as the basis for all aCs of this type.

The first bridge, uniting shungite carbon with other *sp*^2^ aCs was built when comparing vibrational spectra of shungite carbon and one of the synthetic rGOs [5]. It would seem that there was nothing to prevent the idea of NGMs as BSUs from being a fundamental concept for the entire class of *sp*^2^ aCs. However, one day, on the desk of one of my colleagues, I saw two flasks filled with black powder, with manufacturer’s labels pasted on, saying “C 100%”, and with CB624 and CB632 inscriptions indicating SIGMA-ALDRICH MERCK as producers. Both products were synthetic *sp*^2^ aC known as black carbon [6]. The indisputable authority of the manufacturer and the high price of the products did not leave a shadow of doubt about its quality, and none of my arguments based on the quantum nature of the matter and asserting the impossibility of the existence of C 100% nanoscale graphene domains were heard. The only way to solve the problem of the 100% nature of the carbons was to conduct a thorough study of a set of *sp*^2^ aCs using as many available analytical methods as possible. Seven samples listed in Table 1 were selected. All of them were of the maximum carbonization [7,8,9,10,11,12]. The first three are natural *sp*^2^ aCs, the next two are synthetic technical graphenes–rGOs [13] obtained in various ways, and the last two are the above-mentioned black carbons from the SAM. All samples were analyzed by the same set of methods under equal conditions. Three groups of analytic techniques were used: (1) structural one, (2) evaluating the overall atomic composition of the samples, and (3) determining the atomic composition of covalent C–A bonds (A = C, H, O).

The results obtained allowed us to solve the main problem and obtain indisputable evidence that BSUs of *sp*^2^ aCs have a unified nature and are NGMs, the size of whose domains and atomic composition of the relevant necklaces are determined by the chemical history of the amorphics’ origin. The next result showed that all the studied aCs have a multilevel structure based on zero-level BSUs with subsequent aggregation of the units into stacks, globules, and micro- and/or macroscopic agglomerates of higher levels. The third result is that, due to the structural and compositional complexity of *sp*^2^ aCs, analytical testing of samples cannot be single-technique and requires the use of a set of methods. Last, when the multi-technique analytics of *sp*^2^ aCs is mastered, quantum–chemical modeling of the structure, composition, and properties of their BSUs becomes possible. Thus obtained model BSUs open the way to new analytics of the *sp*^2^ aCs that is a virtual one and is based on the Digital Twins concept [14].

The current review presents a consolidated view on these multi-technique analytics applied to a representative set of *sp*^2^ aCs. Since discussed results are too numerous, an atlas-like format was chosen to facilitate the presentation. Although the results, obtained with the participation of the author, make the main contribution to the figures and tables presented below, those obtained in other groups were also taken into account when compiling the review. Nevertheless, the author apologizes in advance for the possible omission of other important results unknown to her and expresses her gratitude for the relevant indications.

## 2. General Characteristics of the Carbon Amorphicity

As mentioned earlier, *sp*^2^ aCs present only a part of aCs that form a large allotropic class of solids, both natural and synthetic. Many books and reviews are devoted to the issue, among which [15,16,17,18,19,20,21,22,23,24,25,26,27,28,29,30,31], but a few, allow the reader to create his/her own idea of the complexity and high science intensity of the touched issue. Natural amorphics are products of nature’s laboratory for geological billion-million-year time. Geologists’ examinations suggest a few classification schemes of carbon species [32,33,34,35]. One of which, presenting a continuous evolution of aC as increasing carbonization rank of pristine carbonaceous masses into ordered crystalline graphite thus exhibiting the main stream of carbon life in nature [36], is shown in Figure 1a. As seen in the figure, a general motion is split into two gloves, the left of which starts with plants and sediments, proceeds through sapropels to brown coals, and later to convenient coals and anthracite, finishing with graphite. As for the right glove, it covers carbonization of pristine gas and distillate oil and proceeds through petroleum and naphthoids, to asphalts, and then to kerites, anthraxolites, and shungites. As in the previous case, graphite is the endpoint. This scheme is related to *sp*^2^ aC that actually dominates in nature [36]. Natural *sp*^3^ aC is not as largely distributed, due to which diamond-like amorphous solids have only recently become top issues of the carbon mineralogy [37,38].

The family of synthetic aCs is quite large, covering species different not only by the carbonization rank, but also by a mixture of *sp*^2^ and *sp*^3^ components. Analogously to natural species, synthetic amorphics were classified as well [36,39,40] and the relevant classification scheme is presented in Figure 1b. Previously ternary, it was completed to a rhombic one to take into account oxygen as another important ingredient [36]. Comparing schemes presented for natural and synthetic aCs, it becomes evident that if natural aCs belong to *sp*^2^ carbon family and are carbonization-rank characterized, synthetic aCs are mainly *sp*^3^-configured solid carbons. The *sp*^2^ group of the synthetic species, characterized by the highest carbonization, takes only a small oval-mapped place. Because of the small amount of *sp*^3^ solid aC in nature, special technologies to produce ta–C, ta–CH (t means tetragonal), and sputtered *sp*^2^ & *sp*^3^ mixed a–C:H products were developed. It is necessary to complement this part of carbon solids by graphite oxide (GO), which corresponds to *sp*^3^ configured carbon with up to ~ 40% of oxygen in the case of complete oxygenation. As for *sp*^2^ synthetic aCs, for a long time they were presented by multi-tonnage industrial production of glassy carbon, covering graphitic, black, activated, and other highly carbonized products [41]. However, the graphene era called to life a new high-tech material—technical graphenes, or rGOs [13], which are the final product of either oxidation-reduction [10,42] or oxidation-thermally-shocked exfoliation [11,43] of nanosize graphite. Two new members of this community are on the way—laser-induced graphene (LIG) manufactured by multiple lasing on cloth, paper, and food [44] and extreme quality flash graphene (FG) [45], closest in ordering to graphite.

Despite aCs having been the object of study and practical use for hundreds of years, only recently have they been considered from the general concept of the solid-state physics [36] basing on fundamentals of the amorphicity of solids accumulated in monographs [3]. The first conceptual issue concerns a considerable degree of amorphous solid ordering that is subdivided into short-range (local) and medium-range ones, the boundary between which passes around a few nm. The second issue is related to the direct interconnection of the solids’ properties and their local ordering due to which establishing a short-range structure has always been the main goal of study. The peak development of this topic of solid-state physics was reached in the third quarter of the previous century. Historically, the most attention has been given to monoatomic Si and Ge, which, it would seem, is quite conducive to our study, since carbon, silicon, and germanium form a common tetrel’s family of the Mendeleev’s table so that a similar behavior could be expected of all the members. As was found, tetrahedrally bonded *sp*^3^ configured atoms form the short-range order of Si and Ge amorphous solids. However, in the case of carbon, as seen in Figure 1b, similar amorphous compositions of carbon are concentrated only near *sp*^3^ corner and are related to the ta–C phase. This internal protest of carbon atoms against the *sp*^3^ hybridization of their electrons in the solid state is confirmed by extremely severe conditions for the formation of diamonds (40 GP and 960–2000 °C) [46]. In contrast to Si and Ge, the main part of the pure C-solids is *sp*^2^-configured. A comprehensive discussion of this peculiarity related to the tetrel’s family members is outside the topic of the current review and can be found in monograph [28]. Thus, monoatomic solid carbon has the unique ability to form amorphous (as well as crystalline) states of two types, characterized by fundamentally different short-range orders presented by either tetrahedral *sp*^3^ groups of bonded atoms or an *sp*^2^ honeycomb network of benzenoid units, graphene domains, thus strongly differentiating *sp*^3^ and *sp*^2^ aCs. Throughout this review, we will mainly talk about *sp*^2^ aCs.

Recent comparative studies [47,48,49,50], taken together with a large pool of individual data, have shown that the general architecture of both natural and synthetic *sp*^2^ aCs is common and can be characterized as a multilevel fractal structure [36,51,52], albeit differing in detail at each level. The structure of the first level is similar in all the cases and is presented by necklaced graphene BSUs introduced earlier. The necklacing plays a decisive role, ensuring the formation and stability of short-range order, on the one hand, and preventing graphitization, on the other [1,2], thus allowing the attribution of the *sp*^2^ aCs origination to the reaction amorphization [3]. The second-level structure is provided with nanometer-thick BSU stacks, which were confidently recorded by X-ray and neutron diffraction structural studies of *sp*^2^ aCs of all types [48]. Once standard, BSU stack pattern forces us to expand the amorphics’ short-range order, completing individual BSUs with their stacks and thus distinguishing *sp*^2^ aCs from routine molecular amorphics [4,53,54]. The third-level structure of the amorphics reliably follows from the porous structure evidently observed experimentally [52,55]. It is constructed from the BSU stacks but the final compositions depend on the stacks’ lateral dimension. When the latter is at the first nanometers level, the composition presents globules of ~10 nm in size, which well correlates with pores, the size of which is first nanometers as well [52,56]. Further aggregation of globules leads to the formation of micro-nanosize agglomerates with pores of tens nm and more [52]. Such a structure is typical to natural *sp*^2^ aCs such as shungite carbon, anthraxolite, anthracite, as well as black carbon coating of diamonds [57], mixed carbon-silica spherical ‘sweets’ [58], black carbon in meteorites [59,60] and metamorphosed polymictic sandstones [61], biogenic carbon, originated from anaerobic oxidation of methane [62], and none of the exclusions have been known so far. Figure 2 presents the evolution of the structure of this kind of *sp*^2^ aC from a single BSU to macroscopic rocks. A schematic structure of a globule, shown in the figure, is designed on the data related to shungite carbon. Molecule C_66_O_4_H_6_ constitutes one of the possible models of the BSUs based on the graphene domain C_66_ [49]. The model is commensurate by size with real BSU and its heteroatom necklace is composed basing on chemical data of the species established in [48]. Combined into four-, five-, and six-layer stacks in accordance with empirical structural data [47,48,49,50], the molecules create a visible picture of the nanoscale globules, further agglomeration of which provides the final nanostructured view of the species presented by the 3D AFM image in the figure. In contrast to natural bodies, synthetic amorphics are characterized by a large dispersion of BSU size from units to tens and/over first hundreds of nanometers. At the low-limit end of the dispersion, their structure is similar to that of natural species described above. At the high-limit end, the BSU size does not prevent from BSU packing in nanosize-thick stacks while the latter laterally extended are further packed in a paper-like structure. The above concept concerning *sp*^2^ aCs structure is based on planar BSUs. The latter are indeed characteristic of real structures, evidence of which are numerous. However, from time-to-time images of natural solids exhibit bent fragments. The issue will be discussed in Section 3.2.

As mentioned earlier, *sp*^2^ aCs do not match any of the known types of disorder that are characteristic for monoatomic covalent solids [3]. Considering this, the *sp*^2^ carbon amorphization can be attributed to enforced fragmentation of graphite [36]. Obviously, fragmented product can be obtained from both the top and bottom. In the first case, it means the disintegration of pristine graphite, while in the second case, it concerns stopping the graphitization of pristine graphene lamellas. There might be various reasons for fragmentation, including mechanical impact, chemical reaction, temperature shock, exposure to hard radiation, etc. However, each fragmentation act is completed with a chemical reaction providing stabilization of necklaces around fragmented graphene domains. Thus formed BSUs, mainly, and their stacks, additionally, ensure the short-range order of the *sp*^2^ aCs. They are characterized by large variety with respect to not only different classes of aCs, but to the same class as well. The variety concerns the BSU size, shape, variation of necklace chemical content, and, most importantly, the local distribution of heteroatoms in the BSU necklaces at fixed atomic percentage on average. Thus, to complete empirical analytics of *sp*^2^ aCs with a virtual one, a large family of NGMs should be associated with each real sample [64]. A single member of the family, shown on the top of Figure 2, is only ‘one snapshot’ of communities related to one of the possible permutations of hydrogen and oxygen atoms in the framing area.

Completing the introduction, a few words should be added about the radical nature of the *sp*^2^ aCs BSUs. The latter is provided with chemical activity of valence uncompensated *sp*^2^ carbon atoms, among which non-terminated edge atoms of graphene domains (these atoms of the NGM C_66_O_4_H_6_ are clearly seen on the top of Figure 2) are the most active. In spite of this, the BSUs remain stable radicals due to the spin-delocalized character of the molecule radicalization provided by the conjugation of unpaired *sp*^2^ electrons over the total number of carbon atoms, nearly degenerated spin-triplet energy gap *E_ST_*, incorporation of heteroatoms (O, N, S) inside benzenoid units or outside the latter [65]. The sleeping activity of the bodies can be stimulated by the additional fragmentation [66], as is in the case of the *sp*^2^ aCs catalytic application [67].

## 3. Structure of *sp*^2^ Amorphous Carbons

### 3.1. Short Foreword

Once started as classical continuous bulks, today’s *sp*^2^ aCs unhesitatingly take their place among nanotechnological materials. Their scientific perception was developing alongside the growing material science, receiving at each turn of the progression a stimulating pulse of a deeper penetration into the understanding of their nature. This development, support, and stimulation would not have been possible without simultaneous progress of instrumental analytical technology. Such a coordinated movement clearly manifested itself to be applied to the structural studies of *sp*^2^ aCs. A perfect historical review on the structure of carbon materials explored by local transmission electron microscopy and global powder diffraction probes [68] presents this growth in the best way. A set of modern techniques, suitable for studies of the structure of nanomaterials, includes atomic force microscopy (AFM), scanning electron (SEM), scanning transmission electron microscopy (STEM), high-resolved transmission electron microscopy (HRTEM), X-ray powder diffraction (XRPD), and thermal neutron powder diffraction (NPD). All these methods have their advantages and disadvantages, and none of them is sufficient to completely disclose the *sp*^2^ aCs structure. The best success is achieved with the joint use of electron microscopy (EM) and powder diffraction (PD) [68]. It is this combination which will be the focus of this review as well.

Structural studies of *sp*^2^ aCs are obviously complicated by the multilevel structure of a real object. Starting from the bottom, the amorphic structure gradually becomes more complex from individual BSUs to BSU stacks, then to globules of aggregated stacks, and finally to micro- and/or macro-scopic agglomerates. None of the analytical methods described above “sees” the structure of the matter in its entirety. Thus, individual BSUs, which are molecular entities, are not atomically seen by any of the methods. Only HRTEM makes it possible to detect the presence of individual BSUs of first nm in size as bar-like images of vertical projections of the areas occupied by carbon atoms. Simultaneously, HRTEM reveals the presence of BSUs’ stack structure thus making possible to estimate the degree of turbostraticity in the latter [69,70]. If the corresponding microscope is equipped with a console that provides fixation of the electron beam diffraction on the sample, then it may determine the interlayer distance *d* inside the stacks of BSUs [13]. In the number of cases related to medium-resolution either TEMs or STEMs, it is possible to reveal well-structured globules of 10–20 nm in size. Images obtained with submicron-resolution EM instruments, as well as with AFM, represent the samples under study in the form of microscopic agglomerates. It should be taken into account that all EM methods deal with a strictly limited part of the object under consideration. All the obtained results are local and may change, sometimes significantly, when passing to another place of the studied sample.

XRPD and NPD are powerful techniques related to samples as a whole. However, they concern the stack structure only, but allow determining not only interlayer distance *d* inside the stacks, but linear dimension *L_a_* of BSUs, that constitute the stacks, as well as thickness of the latter *L_c_* (see the description of the values in Figure 2). Because of significant variability of the sample amorphous structure, the obtained data are statistically averaged and may differ from local ones, obtained microscopically.

Summarizing this short foreword in terms of amorphous state physics, we may state that the modern structural techniques can distinguish short- and middle-range order of solid *sp*^2^ aCs, but in different way. Thus, HRTEM sees the former quite definitely, both as individual BSUs and as stacks of them, but quite locally with respect to the sample body. STEMs of high resolution may visualize the second one as globules of stacks, also locally. Both PD techniques visualize the short-range order in statistically averaged values of the interlayer distanced *d* in stacks, the stack thickness *L*_c_, and linear dimension of BSUs *L*_a_, composing the stacks [69]. In what follows, a representative set of *sp*^2^ aCs will be presented in light of EM an PD techniques applied together. This investigation concerns a set of samples listed in Table 1.

### 3.2. Electron Microscopy

Figure 3 presents a collection of EM results that demonstrate the analytics ability. A comparative view of the images presented in Figure 3a–c allows seeing that microscopic particles of the solids have a common inner structure based on nanosize elements. At the same time, the structures themselves are quite different, apparently evidencing different characteristics of aggregation that precede the solidification. Evidently, the latter occurs differently in nature (cf. Figure 3a,b) while being similar in nature and industry (cf. Figure 3a,c), and again differently when going from industrial production of carbon black to chemically synthesized Ak-rGO and TE-rGO (cf. Figure 3c–e). In the latter case, both GO and rGO often present paper-like solids [42]. At the same time, HRTEM studies of these solids reveal a picture similar to those shown in Figure 3a–c.

When EM, used earlier to study shungite carbon [71,72] in particular, did not allow for establishing the multilevel structure of *sp*^2^ aC unambiguously, today the possibility of using it with different magnifications makes it possible to reliably verify this. This can be traced when looking at Figure 3a,f. As the resolution increases, the structure of microparticles (conditional fourth level of the ShC structure) in Figure 3a is replaced by aggregation of globules with an average size of tens nm (third-level structure) in Figure 3f and then by a set of elements of the first nanometers in size (second- and first-level structure) in Figure 3a. Similar globular images were observed for anthraxolites as well [73,74].

Figure 3g–j present a detailed view of the first two-level structure of shungite carbon [13]. Bars from several fractions of a nanometer to several nanometers long are clearly visible in these images. They are the projections of carbon atom planes oriented almost parallel to the electron beam. It is clearly visible that these structural fragments are grouped into stacks. The distance between fragment planes in the stacks was estimated using a tight connection between the HRTEM image and diffraction of the electron beam providing Fourier diffraction patterns of the former [75]. Thus, the selected region in the HRTEM image in Figure 3j has a Fourier diffraction spot pattern, which corresponds to the interlayer spacing (*d* = 0.34 nm) of a disordered graphite-like material [13].

The atomic planes grouped into stacks of 4–7 layers are clearly visible in Figure 3g–i. As is seen, flat fragments are indeed characteristic of the real structure and are numerous. However, the images also contain bent fragments. Since interlayer spacing *d* for bent fragments is usually 0.34–0.38 nm, any chemical modification of BSUs within their basal planes, which might cause the bending, should be excluded. Accordingly, it was suggested to explain the bending of primary flat BSUs by the existence of various mineral inclusions outside the units. In fact, such impurities accompany ShC formation (from silica micro- and nanoparticles to metal nanoparticles [7]). Proving the suggestion has revealed a unique opportunity of HREELS to study the atomic structure and the elemental composition of the substance under study. The latter can be disclosed when EM is combined with point-like energy dispersive spectral (EDS) analysis. Figure 4 presents an example of the ShC elemental composition mapping with respect to C, O, Si, and Fe constituents. It is seen that only carbon is uniformly distributed throughout the sample, whereas other elements are heterogeneously dispersed. The iron map attracts a particular attention revealing cluster segregation of fractions of a nanometer in size. Similar clusters, containing other metals, are observed as well. Primary BSUs may willingly cover metal nanoparticles and bend, similarly to graphene sheet placed on a heap of gold nanoparticles [76].

### 3.3. X-ray and Neutron Powder Diffraction

X-ray diffraction and elastic neutron scattering are widely applied to determine short-range order parameters of amorphous materials [77,78,79]. Figure 5 presents a typical set of data that provide such structural data relating to *sp*^2^ aCs that concern BSU stacks and BSUs themselves. As is known, diffractogram reflexes Gr (*00l*) are mainly sensitive to the interlayer distance in graphite-like bodies while Gr (*hk0*) ones involve information related to graphite-like structure in the layer plane [68]. As seen in Figure 5a,b, all the plottings are similar and quite scarce, once expectedly concentrated around reflexes Gr (*002*). The abandon richness of NPD spectrograms is usually provided with numerous additional reflexes caused by neutron scattering in crystalline aluminum of a cryostat (see detailed comments in Ref. [80]). Gr (*002*) reflexes are located in the region of 3.3–3.5 Å, which determines interlayer distance *d* along *c* axis between the neighboring layers in graphite thus pointing to undoubted graphite-like stacking of the relevant BSUs. No less important are reflexes Gr (*110*), located in the region of 1.1–1.3 Å, which characterize the size of BSU stacks in lateral directions [69,70]. As seen in the figure, for all the studied *sp*^2^ aCs Gr (*002*) and Gr (*110*), reflexes are shifted (up and down, respectively), as well as considerably broadened, pointing convincingly to a considerable size restriction for the relevant BSUs in both directions.

The broadening of diffraction reflexes is usually attributed to the narrowing of the coherent scattering region (CSR) of a scatterer in the relevant direction [82,83,84]. According to Scherrer’s formula, the full width at half maximum (FWHM) of diffraction peak B and the CSR length $L_{CSR}$ are inversely connected:
(1)LCSR=kBcosΘ.
where λ and *Θ* are the neutron and/or X-ray wavelength and scattering angle while k is a factor depending on the reflex under study [82]. The factor determination is a permanent problem of a quantitative diffraction study of nanosize objects. However, when the study is performed for a set of samples under the same conditions, $L_{CSR}$ can be addressed to the reference one as [80]:(2)$$L_{CSR}^{}=\left(B_{ref}/B_{}\right)\left(\lambda_{}/\lambda_{ref}\right)L_{CSR}^{ref}$$
where, $L_{CSR}^{ref}$, evidently attributed to crystalline graphite, constitutes ~20 nm along both *c* and *a* directions [85].

A comparative view of Gr (*002*) and Gr (*110*) reflexes of *sp*^2^ aCs is presented in Figure 5c–e. As seen in Figure 5c,d, NPD and XRPD reflexes Gr (*002*) are similar. Gr (*110*) reflexes are less intense and quite noisy due to which the usual abandon richness of NPD seriously prevents accurate working in the place while XRPD reflections remain accessible. Table 2 accumulates $L_{c}$ and $L_{a}$ data, determined by applying Equation (2) to the obtained data. As seen in the table, maximum positions, d, of Gr (*002*) reflexes, provided by NPD and XRPD measurements of the studied aCs convincingly evidence that all the samples consist of stacks formed by graphene-like BSUs. The average interlayer distance in the stacks constitutes d = 3.47 ± 0.10 Å, thus remarkably exceeding d in graphite crystals. This was to be expected, since the components of the BSU stacks are not flat one-atom-thick bare graphene domains, but NGMs. The van der Waals thickness of the necklace heteroatoms and their possible deviation from the domain plane is the main reason for the increase in this standard graphite parameter.

## 4. General Atomic Content of *sp*^2^ Amorphous Carbons

### 4.1. Short Foreword

Since BSUs are responsible for the short-range order of *sp*^2^ aCs, it is these bodies that govern the chemical content of the whole solid. Evidently, the presence of BSUs’ necklaces determines the unavoidable heterogeneity of the latter. The necklace contribution into the total atomic content is structure-sensitive. It gradually decreases from a few at % to a negligible amount when the BSU size grows from the first nanometers to micrometers. As seen in Table 2, BSUs of the selected set of *sp*^2^ aCs are nanosize to be proper for heterogeneous chemical analysis. Modern analytic techniques are usually specified for the determination of particular sets of elements. Carbon, hydrogen, and oxygen are the main triad of organic chemistry, so that their dominant presence in the aCs is fully expected. Analyzing the available assortment of analytic techniques, one can make sure that practically all of them are able to fix carbon. As for hydrogen and oxygen, the techniques are distinctly divided into {C,H}, {C,O}, and {C,H,O} groups. Following this division, we subordinate further description to this grouping.

### 4.2. {CH} Analytics

Analytic techniques are qualitative, quantitative, or mixed. The first are those based on chemical or physical properties that reveal a strong dependence on the chemical entity under study. In the case of *sp*^2^ aCs, combustion and incoherent inelastic neutron scattering (IINS) of hydrocarbons are widely used for detecting hydrogen. The former lays the foundation of differential thermal analysis (DTA) and differential scanning calorimetry (DSC) [86], the latter, less popular, presents inelastic incoherent neutron scattering (IINS) vibrational spectra [87].

The DTA/DSC analysis mainly concerns the *brutto* content of sample carbon, so called *fixed carbon* following the equation
% Fixed Carbon= 100 − (% Adsorbed Water +% Ash + % Volatile Matter)(3)

The fixed carbon is definitely not carbon-pure and must at least include hydrogen and oxygen components, which are highly combustible constituents in addition to carbon. As occurred, DSC thermograms of a set of *sp*^2^ aCs, obtained at the same conditions, are quite different [48] while the carbonization of all the samples is over 90 %. Figure 6a presents the data collection, which evidently reveals changes in the temperature of the process onset *T_ost_*, from 370 °C to 720 °C. Thus, as seen in the figure, *T_ost_* of graphite is the largest. In contrast, on the opposite side of the observed series of the DSC thermograms, there is one related to a lab-produced rGO [88], *T_ost_* for which constitutes 370 °C. As is known, rGOs, both natural and synthetic, usually contain a few wt% of hydrogen. Accordingly, the presented DSC series may evidence a gradual depletion of hydrogen content in the *sp*^2^ aCs series from ShC to CB624 and graphite. In this case, the *T_ost_* below 720 °C obviously indicates the hydrogen presence in the solid, therewith, the lower the *T_ost_*. DTA/DSC are fully qualitative techniques. However, a comparative study of a set of samples under equal conditions allows for disclosing a solids’ comparable series following the decrease in hydrogen content, such as the following:rGO > ShC > AntX > CB632 > CB624 > Gr.

In its turn, protium is a peculiar hydrogen isotope of the largest cross-section for the IINS among all the elements, making the relevant techniques highly protium sensitive. The abundance of a real hydrogen isotope family with protium readdresses the statement to chemical hydrogen [87]. Thus, water is willingly adsorbed with both natural and synthetic *sp*^2^ aCs, which is evidently caused by their porous structure and hydrophilic nature of the BSUs necklaces that cover inner walls of pores. Expectedly, Figure 6b presents a strong reaction of IINS spectra on the gradual hydration of CB632. As shown in Figure 6c–e, each as-prepared *sp*^2^ aCs involves a large quantity of adsorbed water and must be freed from it before analysis for the hydrogen content. In their turn, dry samples of the latter still remarkably scatter neutrons, thus exhibiting the hydrogen presence in the solids related to the necklaces of their BSUs. So far, none of IINS’ quantitative analyzes for hydrogen in *sp*^2^ aCs has been developed, leaving it mainly qualitative. However, when a study of a solids’ set under equal conditions is possible, a comparative analysis of the obtained IINS spectra, as is in the case presented for dry samples in Figure 6c–e, allows a confident conclusion that the hydrogen content in the AntX and ShC BSUs is comparable, while that one for the CB632 is much lower. The relevant comparative series of samples looks like the following:ShC ≈ AntX > CB632 > Gr.

This tendency is quite similar to the one followed from the DSC data in Figure 6a, while not identical. Apparently, it may be caused by the fact that DSC results were obtained for as-prepared samples that could not be fully freed from the adsorbed water.

In contrast to the above techniques, a combustion-based elemental analysis (EA) can provide a qualitative hydrogen analysis of *sp*^2^ aCs. Standard *CHNS* EA involves determination of carbon and hydrogen contents supplemented with the detection of nitrogen and sulfur. It does not determine oxygen content directly and the relevant data are just residual content of 100 wt% samples mass after excluding all other contributions. The hydrogen data related to the solids listed in Table 1 are presented in Table 3. The comparative sample series looks as follows:AntC > AntX > TE-rGO ≈ Ak-rGO > ShC > CB632 > CB624.

There is much common and different in the triad of comparative series evidencing that obtaining exact values of hydrogen content even in the same samples of *sp*^2^ aCs is hardly possible. None of the techniques are free from particular limitations related to both the performance and interpretation of the results obtained. However, all of them are highly important and useful.

### 4.3. {CO} Analytics

Since *CHNS* EA does not provide the determination of the oxygen content, EDS and X-rays photoelectron spectroscopy (XPS) take on the main role. Both techniques are qualitative and quantitative, thus detecting the oxygen presence in samples and determining its atomic percentage. Figure 7 presents a general view on these techniques’ abilities. Qualitative EDS, known as EDS mapping, allows for registering of the distribution of a selected element over the sample by monitoring its monochromatic X-ray emission [89]. Typical EDS maps for ShC are presented in Figure 7c–e. As seen in Figure 7e, oxygen is largely presented in the solid. The quantitative EDS analysis (Figure 7a) concerns a comparison of the intensity of element-wave-dependent X-ray emission with that of a standard reference element registered at identical conditions. A characteristic feature of EDS is the localized character of its data. The measurements are usually performed at inclusion-free sites controlled by EDS maps, similar to the O-map in Figure 7e. The data should be averaged over those determined at several different places for each sample. As shown in Figure 7f, the data related to different places of the sample scatter quite remarkably. Figure 7a presents a collection of EDS spectra of a set of dry *sp*^2^ aCs samples while partitioned chemical components are presented by spherical diagrams. As seen in the figure, the emission spectra are presented by main signals related to carbon, the amount of which covers the region of 95.05÷97.44 wt%, and oxygen of 1.7 ÷ 3.3 wt%. In addition, a variable set of minor impurities is usually obtained. EDS analysis confirms that *sp*^2^ aCs are not carbon-pure but involve a few percentages of oxygen and ~1.5 wt% of minor impurities.

In contrast to EDS, XPS data are statistically space-averaged. A general panorama of the survey XPS spectra of a set of *sp*^2^ aCs is presented in Figure 7b. It is usually assumed that a high degree of vacuum provides the release of the studied samples from the adsorbed water. Quantification of atomic content is provided using sensitivity factors from the elemental library of CasaXPS [90]. The XPS spectra of the studied samples look quite similar, evidencing the C1s and O1s spectra of the main chemical components and allowing for the evaluation of atomic percentage of the observed elements by standard technique. The obtained data are given in Table 3. As seen in the table, all the studied *sp*^2^ aCs contain a considerable amount of oxygen.

Concluding the chemical testing, is necessary to pay attention to the following: all the methods allow for the direct determination of carbon content. The dispersion of the latter occurs quite largely and constitutes 95.2 ÷ 88.5 wt% for ShC, 95.6 ÷ 89.5 wt% for AntX, and 97.94 ÷ 90.7 wt% and 99.67 ÷ 93.1 wt% for CB632 and CB624, etc., respectively. This feature clearly evidences the multi-elemental character of the species, on the one hand, and the different sensitivity of the used analytical techniques to chemical elements that accompany carbon in the studied samples, on the other. It is important to note as well that all the reference carbons unavoidably used in testing are not pure carbons as well. Oxygen is the main impurity, which may point to a heightened propensity of pure carbons to oxygenation. Hydrogen is the next contributor. In addition, the hydrogen weight content is comparable with that of other minor impurities involving sulfur, nitrogen, chlorine, silicon, and different metals. The role of the latter is particularly important in the geochemistry of carbon [91,92] or the engineering technology of highly carbonized products [93]. Once concentrated on disclosing the BSU atomic structure, we shall restrict ourselves in what follows by the consideration of carbon–hydrogen–oxygen triad that present necklaced graphene oxyhydrides. The latter are covalent species characterized by a large variety of covalent bonds between the triad elements, attribution of which to particular atomic pairs is provided with the {*CHO*} analytics of *sp*^2^ aCs.

### 4.4. {CHO} Analytics

There are many methods for the direct or indirect determination of the composition of covalent bonds. As applied to *sp*^2^ aCs, the most successful are vibrational spectroscopy in the first case and XPS in the second. Vibrational spectroscopy includes a wide range of different methods based on the inelastic scattering of elementary particles (photons, neutrons, electrons, γ particles, neutrinos, etc.). Each of these methods is characterized by its own way of extracting information about the covalent bonds of a scattering substance from the corresponding spectra. Methods for the indirect study of the bonds are based on the reaction of the binding energy of atoms to the presence of other atoms bound to them, the most prominent representative of which is XPS. This method is widely used in the analytics of *sp*^2^ aCs, and that is why we will begin the review of {*CHO*} analytics with it.

#### 4.4.1. XPS Analysis

In contrast to IINS, which is a ‘hydrogen tool’ and which provides a reliable qualitative test of the hydrogen presence in *sp*^2^ aCs, XPS in general, and its O1s spectra in particular, is considered as the main ‘oxygen tool’ of the solids. The determination of O/C content, exhibited in Figure 7b, is usually followed with a detailed analysis of covalent bonds, formed by the element within the bodies’ BSU necklaces. Figure 8 presents a general view on a massive set of XPS results, for a selected set of *sp*^2^ aCs listed in Table 1. As mentioned earlier, XPS analysis of chemical bonding is based on two fundamental facts: (1) the atom-dependent value of binding energy ($BE_{A}$) of electrons on internal orbits of the *i*–th atom, $BE_{iA}$; and (2) the $BE_{iA}$ value’s dependence on the *i*–th atom neighboring, which allows disclosing the type of chemical bonding of the atom to other ones. Both features are clearly visible in the case of *sp*^2^ aCs carbon (C1s) and oxygen (O1s) XPS spectra shown in Figure 8a. The tailing of C1s spectra and broad O1s ones are commonly associated with the multi-variable response of carbon and oxygen atoms to their surroundings. For a long time, the relevant XPS spectra have been analyzed in terms of the ‘four peaks’ approximation that involves groups C–C, C–O, C=O, and COO of C1s spectra and C–O, C=O, C(=O)O, and O=C(O) groups of O1s spectra [94,95,96,97,98,99,100,101,102,103,104]. In the latter case, the corresponding components are usually associated with the simplest oxygen containing groups (OCGs) such as hydroxyls, epoxides, carbonyls, and carboxyls. The approach resulted from the extended XPS study of numerous different polymers of known structures, which laid the foundation for atomic group assignment to characteristic XPS peaks [105]. However, this approach happened to conflict with IR absorption studies, which revealed much more complicated OCGs, including benzenoid heterocycles such as ketones and quinones, cyclic ethers, lactones and acid anhydrides, furan and pyrans, as well as hydroxypyrans and so on [106,107,108,109,110,111]. Evident questions were raised in the case of XPS [112], the answers to which were collected in a profound review [113]. Thus, a ‘five peaks’ approach was suggested to decompose O1s spectra, which was realized with respect to spectra shown in Figure 8a [48,49]. The modified asymmetric Voigt functions [114] were adopted to separate waveforms of XPS spectra following the group binding energies listed in Table 4. A full collection of the decomposed data is presented in Figure 8b, while a comparative view of them relating to natural, synthetic, and industrial *sp*^2^ aCs is shown in Figure 8c–e, respectively. Thus, the revealed OCGs related to the BSUs of a selected set of *sp*^2^ aCs are summarized in Table 5.

#### 4.4.2. IINS Analysis

A high efficiency of IINS spectroscopy as an H-tool is clearly seen in Figure 6. Original IINS spectra depend on the instrumentational peculiarities of the spectral devices in use. Accordingly, the spectra recorded either on the NERA spectrometer of the high flux pulsed IBR-2 reactor of the Frank Laboratory of Neutron Physics of JINR, or on the TFXA spectrometer at the ISIS pulsed-neutron source, Rutherford Appleton Laboratory, and on IN6 spectrometer of the ILL should be and, actually are, different. However, evidently, common information related to the vibrations of a scatterer involved in the action can be provided by converting the original data to the generalized density of vibrational states (GVDOS) by the relevant standard programs [87]. A collection of such GVDOS spectra, related to a set of *sp*^2^ aCs listed in Table 1 [49], is shown in Figure 9. The spectra concern dry samples after removing adsorbed water. They are provided with both direct scattering from hydrogen atoms, chemically bound with the edge atoms of the BSUs’ graphene domains, as well as with enhanced scattering from the carbon atoms of the relevant graphene domains due to the ‘riding effect’. The latter is caused by the contribution of hydrogen atoms to the eigenvectors of vibrations related to carbon atoms through the BSU molecules (see detailed discussion of both effects in [5]). The spectra are fine-structured enough to be attributed to particular vibrations, thus establishing the chemical bonding of hydrogen atoms in the solids [87]. As is typical for any vibrational spectroscopy of molecules, group frequencies (GFs) lay the foundation of the covalent bonds analysis [115]. Presented in Table 6, GFs correspond to atomic bonding inside (C,X) groups (X=C,H) [49].

The GVDOS spectra in Figure 9 are definitely divided into two parts. Thus, all the spectra in Figure 9a have much in common in the regions of 960 cm^−1^, 800 cm^−1^, and 600 cm^−1^ in spite of being recorded on different spectrometers, namely, ShC and AnthX on NERA [116] and AnthC on TFXA [117]. Characteristic spectral features are clearly visible in all the spectra, thus indicating a similar involvement of hydrogen atoms in the scattering. Table 6 accumulates group frequencies widely used in the analysis of vibrational spectra of hydrocarbons [118]. According to the table, first two features are attributed to the *in*- and *out-of-plane* bending vibrations of *sp*^2^ C–H bonds of methyne groups, while the third one represents *ip* bendings of carbon atoms of the benzenoid units of the BSU graphene domains enhanced by the ‘riding effect’. Therefore, the IINS study of natural *sp*^2^ aCs reveals that, in addition to the similarity in molecular spatial structure of their BSUs, hydrogen atoms in their necklaces predominantly form sets of methyne *sp*^2^C–H bonds. An intense peak below 100 cm^−1^ in the AnthC spectrum indicates that besides *sp*^2^C–H bendings, torsions of hydroxyls take part in the scattering. A further support of the latter suggestion is given in Section 5.3.2.
nanomaterials-13-01648-t006_Table 6Table 6Group frequencies of aromatic molecules required for the hydrogen-content analysis of vibrational spectra of *sp*^2^ amorphous carbons.Spectral Region, cm^−1^Group Frequencies ^1^(C, C) ^2^(C, H_1_) ^2^(C, CH_2_) ^3^(C, CH_3_) ^4^400–700404 δ *op* C–C–C606 δ *ip* C–C–C -711 ρ CH_2_
210 *r* CH_3_344 δ CH_3_700–1200707 C–C–C puckering993 ring breathing1010 δ C–C–C trigonal 673 δ *op* in phase846 δ *op*, C_6_ libration967 δ *op*990 δ *op*, trigonal1037 δ *ip*1146 δ *ip*, trigonal1178 δ *ip*948 ρ CH_2_900 ν C–CH_3_1041 ρ CH_3_1200–16001309 ν C–C Kekule,1482 ν C–C1599 ν C–C 1350 δ *ip* in phase 1409 δ internal CH_2_
1333 δ CH_3_1486 δ internal CH_3_
2800–3200-3056 ν C–H3057 ν trigonal C–H3064 ν C–H3073 ν in phase C–H3114 ν CH_2_2950 ν CH_3_^1^ Greek symbols ν, δ, ρ, *r*, τ mark stretching, bending, rocking, rotational, and torsion modes, respectively; ^2^ GFs notifications of fundamental vibrations of benzene molecule [118]; ^3^ GFs notifications of fundamental vibrations of benzyl radical [119,120]. Hereinafter, GFs, additional to the benzene pool of vibrations, will be shown only; ^4^ GFs notifications of fundamental vibrations of toluene [121,122].

In contrast to natural amorphics, their synthetic and industrial analogues are characterized by significantly different GVDOS spectra, as seen in Figure 9b. A comprehensive analysis of the spectra [49] showed that the formation of methylene *sp*^2^C–CH_2_ bonds in the BSU necklaces of the solid was the main motive of the H/C bonding in TE-rGO, while the formation of *sp*^2^C–CH_3_ methyls dominates in the Ak-rGO solid. In both cases, the chemical bonding of the BSU necklaces was a direct consequence of chemical reactions that accompanied the reduction of parental graphene oxide [49]. As for the spectrum of industrial CB632, it is very weak while that of CB624 cannot be distinguished over the background at all. This feature points to a practical absence of hydrogen in their BSU necklaces and is evidently connected with the high-temperature pyrolytic conditions of the solids production, which does not maintain the existence of hydrogen-containing radicals in the chemical surrounding. The hydrogen content in CB632 is at the limit of the technique sensitivity due to which only a sharp feature below 100 cm^−1^ marks the presence of hydrogen atoms in the species BSU necklace. Apparently, it may be attributed to the presence of hydroxypyran in the area.

#### 4.4.3. IR Absorption Analysis

Organic chemistry is unthinkable without the widespread use of IR spectroscopy as the main analytical technique. The molecular nature of BSUs of *sp*^2^ aCs undoubtedly puts the latter on par with other objects in the field. The special significance of this type of analysis of *sp*^2^ aCs became apparent after it was found that the heteroatoms of the BSU necklaces, chemically bonded to the graphene domain, make the main contribution to the solids IR absorption spectra [64]. This feature will be discussed in more detail in Section 5.3.2. FTIR and DRIFT techniques are usually used to record the IR absorption spectra of the solids.

DRIFT spectra of *sp*^2^ aCs listed in Table 1 are presented in the top panels of Figure 10. They are combined in groups related to natural, synthetic, and industrial solids. A large variety of the spectra shape convincingly evidence a drastic variation of the chemical content of BSU necklaces of the solids. Evidently, to proceed with the spectra features assignment, a list of GFs related to the (C, O) bonding is needed. Such a list, based on the assignment of frequencies in the experimental spectra of graphene oxides [106,109,113,123] and extended calculations [64,124], is given in Table 7. As seen in the table, practically each spectral region of the spectra is quite ambiguous, which complicates the assignment. However, a combined analysis of IR and XPS spectra greatly facilitates the job. The bottom panels in Figure 10 present the distribution of the intensity of the Voigt fitting function (VFF) peaks (see Figure 8b) over group binding energies related to oxygen O1s state. The relevant GBEs are listed in Table 4.

A joint DRIFT-XPS analysis of the studied *sp*^2^ aCs shows that the features of these spectra can be described by taking into account the hydrogen and oxygen components of the BSUs’ necklaces only [49]. As for the former, it dominates in natural aCs (Figure 10a), and is still significant, but is much less intense in synthetic ones (Figure 10b) and is practically absent in industrial carbon blacks, approaching nil for CB624 (Figure 10c). This tendency is clearly evidenced with a considerable decrease in vertical scales in the figures. In full agreement with the results of the IINS.

Analysis, the main contribution is provided with *ip* and *op* bendings of methyne, methylene, and methyl units. It should be noted as well that the hydrogen contribution to DRIFT spectra of natural solids is three–five times more than that of synthetic ones, while the hydrogen contents of the bodies (see Table 3) are quite comparable. This feature evidences that hydrogen atoms, directly attached the graphene domain edge atoms in the former case via *sp*^2^C–H chemical bonds, are optically much more active than those ones involved in either methylene or methyl units via *sp*^2^C –CH_2_ and *sp*^2^C –CH_3_ bonds, once distant from the BSU domains. This issue was observed in virtual IR spectra of necklaced graphene molecules as well [64,125].

As seen in Figure 10, the oxygen component of the studied solids varies quite drastically, while remaining similar in the samples of each of the three types. Thus, a joint DRIFT-XPS analysis, based on the GBEs and GFs listed in Table 4 and Table 7, allowed for concluding of the following [49]: the oxygen component of the BSU necklaces of ShC may mainly involve carbonyls *sp*^2^C=O and *o*-quinone O=*sp*^2^C–*sp*^2^C=O bonds. That one of AnthX is of lactone character consisting of C=OOC (lactone) and pairs of lactones, as well as of aggregated cyclic ether with lactone. The oxygen content of AnthC is the richest, involving carboxyls *sp*^2^C–COOH, cyclic ethers, aggregated cyclic ethers, pyran, and hydroxypyran. Aggregated cyclic ethers, aggregated cyclic ether with lactone, and lactones are common for both Ak-rGO and TE-rGO, once completed with lactone pairs in the former case and with hydroxypyrans in the second. Similarly, C–O–C in cyclic ether and aggregated cyclic ethers with lactone form the ground of CB632 and CB624, completing the former case with lactones and/or hydroxypyrans. Obtained data led to the foundation of molecular models suggested for BSUs of the studied *sp*^2^ aCs that are discussed in Section 5.2.
nanomaterials-13-01648-t007_Table 7Table 7Group frequencies required for the oxygen-content analysis of BSUs of *sp*^2^ amorphous carbons ^1^, cm^−1^.300–10001000–12001200–13001300–15001500–16001600–17001800–19002600–30003000–3600*δ op*^2^,*δ ip*^3^,τ  C–OH*sp*^2^**C**–**O–C**and*sp*^2^**C**–**O**H*δ op sp*^2^**C–C–C**
^4^*δ ip*,puckering,ring breathing,*δ* trigonal*sp*^2^**C–C–C**
^4^,collective vibrations of graphene domainatoms ^5^*ν sp*^2^**C–O–C** in cyclic ether, aggregated cyclic ether and acid anhydride, *ν sp*^2^**C**–**O**H,in lactone, hydroxyl pyranand acid anhydride *δ ip**sp*^2^**C**–**O**H,*ν sp*^2^**C–O–C** in cyclic etherand acid anhydride*δ ip***O–C=**Oin acid anhydride*δ ip**sp*^2^**C**–**O**H,*ν sp*^2^**C–C***ν sp*^2^**C=O** inacid anhydride and lactone, aggregated cyclic ether with lactone pair,pairs of lactones*ν sp*^2^**C=O**in*o*-quinone, COOH*ν sp*^3^C–**O–H** in COOH*ν sp*^3^**C–H***ν sp*^3^C–**O–H**^1^ Greek symbols τ,  δ and ν mark the molecule torsions, bendings and stretchings, respectively; ^2^ Out-of-plane bendings; ^3^ In plane bendings; ^4^ Benzene molecule data [118]; ^5^ Virtual data for nanographene [126].

#### 4.4.4. Raman Scattering Analysis

Raman scattering analysis has become a champion among a variety of analytic techniques applied to graphene-like materials, not excluding *sp*^2^ aCs. No articles can be published without referring to the Raman spectra of such materials. This extraordinary situation is caused by the evident exclusiveness of the spectra, a general view of which for a set of *sp*^2^ aCs listed in Table 1 is presented in Figure 11. As seen in the figure, the spectra of samples, characterized by different short-range orders and chemical contents of the relevant BSUs, are practically identical by shape consisting of three characteristic regions named D, G, and 2D [127,128]. Following the assignment suggested for Raman spectra of crystalline graphite [129], the doublet of D and G bands corresponds to a one-phonon contribution, while 2D marks the region of two-phonon events.

These one-phonon and two-phonon parts of the Raman spectra in Figure 11 are not exactly identical, as shown by detailed consideration of the spectra structure [50]. However, a characteristic doublet D–G remains highly universal. For a long time, there has not been a convincing explanation for both the deep similarity of the spectra as a whole and the exclusive character of the D–G doublet. Starting from the spectra of crystalline graphite and graphene, consisting of two narrow bands G and 2D, spectroscopists looking for the band broadening and the appearance of D were divided into two groups. The first group members tried to explain the features remaining in the framework of crystal spectrum concepts. This is how the idea arose about the defective origin of the D-band [129,130] and about the double-resonance nature of the 2D one [131,132]. An exclusive role to two parameters of the spectra, namely the ratio of the intensity of the D and G bands, IDIG, and the corresponding bands half-widths, Δω, was given to characterize the size and defect structures of the relevant graphene domains. Established theoretically for graphene crystal, this relationship was transferred to the molecular BSUs of amorphous substances by default [133,134]. However, as was shown lately [50], such a transfer turned out to be incompetent, which, nevertheless, has not stopped the efforts of the “theoretical description of the defectiveness” of the studied BUSs until now.

The molecular nature of *sp*^2^ aCs underlies the position of scholars of the second group [135,136,137,138,139,140,141]. Information presented in previous sections of this article convincingly proves the issue, which makes all the crystal-based theoretical approach practically not applicable. It was shown that the polarizability tensor of a polyaromatic hydrocarbon, to which BSUs evidently belong, depends on the dynamic characteristics of the multimode vibrational spectrum. As occurred, the main contribution to the intensity of the spectra is made by *sp*^2^C–C stretchings, due to which the observed D–G–2D set of bands is mainly characteristic of the network of *sp*^2^C–C bonds. This is the first reason explaining the similarity of Raman spectra of different *sp*^2^ aCs, since similar graphene domains constitute the main atomic part of the relevant BUSs. The second reason concerns the *sp*^2^C–C stretching modes, which are responsible for the bands. It was found that the G-band is originated from the *e_2g_* vibration of benzene, while the modes responsible for the D-band come from the *e_1u_* mode of the molecule. The former provide simultaneous in-plane stretchings of all *sp*^2^C–C bonds, while the latter concern both stretching and contraction of these bonds when carbon atoms move out-of-plane. A final decision of the problem has been obtained recently in the framework of virtual vibrational spectrometry [64]. As occurred, D-bands are, actually, caused by out-of-plane *sp*^2^C–C stretchings that provide formation of a dynamically stimulated *sp*^3^C–C bond between the adjacent layers of BSUs’ stacks, thus being a characteristic test of a particular short-range order structure of the amorphics. Intensity of the D-band increases when the number of layers grows up to 4–5 and then markedly slows down when the total thickness of stacks exceeds ~15 nm [142]. As for 2D region, the same out-of-plane *sp*^3^C–C stretchings are responsible for the features, thus revealing the strongest anharmonicity among other vibrations.

A particular doublet pattern of the spectra strongly depends on the linear size of the BSU and is transformed from a broad-band-doublet one to a narrow-band one with drastic domination of the G-band when the size of graphene domain exceeds the free path of graphene optical phonons $L_{ph}$ ~ 15 nm [143]. Therefore, the characteristic D–G doublet structure of the NGMs’ Raman spectra is of structural origin, evidencing the stacked nature of the corresponding solid structure. Thus, the analysis of the structural and chemical content data, particularly, of the Raman spectra, of the cosmogenic carbon brought to Earth by the Chelyabinsk meteorite [60,144], leads to a confident conclusion about the graphite-like stacked structure of this carbon with a stack thickness of ~ 10–15 nm and more.

## 5. Virtual Analytics of Necklaced Graphene Molecules

### 5.1. Short Foreword

Analysis of *sp*^2^ aCs, performed using the analytic techniques described above, undoubtedly establishes the molecular nature of these solids, determines the solid-forming molecular BSUs as flattened necklaced-graphene type, estimates the average statistical linear size of them and the parameters of their primary stacking, and determines the atomic content of the BSUs’ compositions. However, all these methods taken together do not allow us to describe BSUs using atomic structural forms such as C_x_H_y_O_z_(Imp)_minor_, so familiar for material science, beyond the general idea that they are graphene domains in a necklace of heteroatoms. Obviously, the numbers *x, y, z, minor* represent the realization of the chemical composition of substances, analytically determined as weight or atomic percentage, in atomic structural form. Taking into account the statistically averaged nature of the determined structural and compositional data (cf. Table 2 and Table 3), the high complexity of constructing such a formula becomes evident. The first step towards solving this problem is the reduction of statistically averaged empirical data to point ones. The second step requires determining the formula-generating element and setting the absolute number of its atoms. In the case of *sp*^2^ aCs, carbon atoms of number *x* evidently play the role. Further steps depend on which part of the BSUs is covered by this number. In the simplest case, it refers only to the graphene domain, due to which *y, z* and *minor* numbers determine the heteroatoms located in the BSU’s necklace. If the heteroatom necklace includes carbon atoms such as those involved in carboxyl, methylene and methyl groups, or domain atoms are partially replaced by heteroatoms as in the case of, say, aggregated ethers and lactones, these numbers are determined by successive approximation. Naturally, chemical formulas C_x_H_y_O_z_(Imp)_minor_, constructed in this way and related to the same set of empirical chemical-content data, depend on the number of carbon atoms, i.e., on the size of graphene domains.

However, the presence of chemical formula does not allow for visualizing the BSU atomic structure. The problem is that a huge number of various-shaped graphene domains correspond to a given number of carbon atoms *x*. In its turn, a different number of edge atoms, subjected to targeting with heteroatoms, characterize shape-different domains. Moreover, the arrangement of heteroatoms over a set of the domain edge atoms is a difficult problem due to the large variability of multi-targeting, which is complicated by the high radicalization of these carbon atoms [145]. Therefore, in view of a practically infinite number of potential structural variations, none of the molecular structure of the C_x_H_y_O_z_(Imp)_minor_ chemical content, either drawn voluntarily by pen or designed by following particular algorithms, can take the place of an exact BSU view and is only one of a great number of possible ‘snapshots’. Once so restricted with respect to real structures, the constructed BSUs are nevertheless useful, being the only structural images that allow for distinguishing the BSUs of different *sp*^2^ aCs at the atomic level, immersing into the solids chemistry, exhibiting alive and silent chemical problems associated with the BSUs, disclosing the grounds and potentiality of different applications of *sp*^2^ aCs, and so forth. Moreover, starting as molecular models of real solids, these virtual BSUs are easily transformed into Digital Twins [5,146] laying the foundation for reliable virtual analytics of the solids. Virtual vibrational spectrometry has so far been the first technique for the latter.

### 5.2. Molecular Models and Digital Twins of BSUs of sp^2^ Amorphous Carbons

In due course of the comprehensive analysis of the set of *sp*^2^ aCs listed in Table 1 presented in the previous sections, a sufficient amount of data were obtained to enable the construction of BSUs models of the studied solids. Following general instructions, described in the previous section, rectangular graphene domains (5, 5)NGr and (9, 9)NGr with linear dimensions of 1.12 × 1.22 nm^2^ and 1.97 × 2.20 nm^2^, consisting of 66 and 190 carbon atoms, respectively, were chosen for modeling. According to the data in Table 2, the first domain is commensurate with the BSUs of natural solids and CB632, while the second is much bigger to be closer in size to the BSUs of CB624 and synthetic solids. Since the {*CHO*} triad evidently dominates in the chemical content in Table 3, we restricted ourselves with these elements only. Associated with the two domains mentioned above, the relevant BSU models listed in Table 8 are presented in Figure 12a–g. We remind readers that the structures represent only instantaneous snapshots of numerous potential configurations of the same composition for each of the solids. It turned out that the analysis of numerical data on the chemical composition of BSUs from the viewpoint of XPS only is insufficient for a narrow choice of suitable structures [48]. It was necessary to plug DRIFT absorption spectra as well, which made it possible to significantly narrow the choice of structures to the set shown in the figure. A detailed discussion of building these models can be found in [49]. All structures are stable radicals [65] with temporarily quenched chemical activity. In the same way, BSU models can be built, which include, in addition to the triad of main components, minor impurities of nitrogen and sulfur [66] (see Figure 12h–j). Evidently, these models, although not being exact imprints of the structure, make it possible to deal with *sp*^2^ aCs with open eyes, understanding what the commonality and difference between natural amorphics are, and how the latter differ from synthetic and industrial ones. The difference in their behavior in catalysis [66], optics [147], medicine [148], etc. has the opportunity to become clearer.

This first attempt at building empirically based models of NGMs opened up the possibility for a new direction of analytical research of *sp*^2^ aCs aimed at moving from substances based on BSUs with an empirically unattainable structure, to those built from the NGMs of voluntary modified structure. Thus, going from individually different while not exactly determined empirical BSUs to a class of NGMs, undoubtedly having properties common to the whole class. This transition is possible within a new modeling concept known as the Digital Twins (DTs) [5,146]. The design of DTs is now subordinated not to the reproduction of the structure of empirical BSUs, but to respond to a series of questions aimed at elucidating one or another feature of the NGM class. In fact, when applied to a large set of NGMs [64,124,125], the approach has revealed many of the molecular commonalities, which are of practical interest, and which form the grounds of virtual analytics of *sp*^2^ aCs. First, this concerns virtual vibrational spectrometry that happened to be highly adaptive to a new analytic role.

### 5.3. Virtual Vibrational Analytics of sp^2^ Amorphous Carbon

In molecular science the DT concept can be schematically presented as the following:*Digital twins* → *Virtual device* → *IT product*.

This scheme connects three constituents of the approach [15]. Here, DTs are the molecular models under study, virtual device is a carrier of a selected software, and IT product covers a large set of computational results related to the DTs under different actions in the light of the soft explored. The quality of the IT product depends highly on how broadly and deeply the designed DTs cover all the knowledge concerning the object under consideration and how adequate the virtual device is to the peculiarities of this object. Applying virtual vibrational spectrometry, the virtual device is a virtual spectrometer. It should not contradict with the object nature and will perform calculations providing for the establishing of equilibrium structure of the designed DTs and obtaining their spectra of IR absorption and Raman scattering related to 3*N–6* vibrational modes. Virtual spectrometers differ by the software in grounds and are of HF (Hartree–Fock), DFT (density of functionals) or MD (molecular dynamics) type, since only semi-empirical programs based on the mentioned approximations can cope with the large volume of cumbersome calculations that are needed. As for quantum approaches, the radical nature of *sp*^2^ aCs, caused by the open-shell electronic systems of their BSUs, forced us to abandon DFT- and MD-based softwares and pay particular attention to programs based on the unrestricted Hartree–Fock approximation. All the results illustrated below were obtained by using the virtual vibrational spectrometer HF Spectrodyn [149].

#### 5.3.1. IINS Virtual Analytics

A standard model aimed at ‘reading out’ the empirical vibrational spectra of *sp*^2^ aCs started far before the DT concept was introduced. The main problem preventing the issue from developing concerns the difficulty in building proper models. The situation changed appreciably when the first proposals to use NGMs as such models appeared. The models were initially applied for virtual IINS analytics of *sp*^2^ aCs. Its main goal is to reproduce the solids behavior in the relevant empirical experiments. One-phonon GVDOS was the subject of calculations [87]. Two virtual experiments have been performed so far. The first mainly concerned adsorbed water in CB632, the empirical IINS spectrum of which is shown in Figure 6b. A simplified BSU model, slightly different from the more detailed model that is shown in Figure 12d, served as a carbon substrate immersed in the cloud of water molecules [2]. The second is related to a comparative analysis of empirical IINS spectra of *sp*^2^ aCs presented in Figure 9 using the first BSUs models, similar to those presented in Figure 12 [47].

As seen in Figure 13, the obtained virtual GVDOS is in line with the empirical data related to adsorbed water. In the second case, a set of virtual GVDOSs presented in Figure 14 quite satisfactorily reproduced empirical data in the part related to the dependence of the spectra intensity on the hydrogen content of the relevant BSUs. Nearly linear dependence of the total spectrum intensity on the number of hydrogen atoms in the necklaces of the model BSUs was obtained [47], which follows the tendency of the hydrogen content of real *sp*^2^ aCs listed in Table 3. A large width of the real IINS spectra does not allow for performing a more refined search of the BSUs models suitable for the spectra description [49].

#### 5.3.2. IR Absorption Virtual Analytics

The germination of virtual IR absorption analytics of *sp*^2^ aCs has coincided with the beginning of transition of standard modeling of their individual molecular BSUs to the DT concept that concerns the class of NGMs [15,64,124,125]. A set of relatively reliable structural BSU models made it possible to penetrate their chemistry and begin to understand the main features of NGMs. In turn, the latter circumstance opened the way to recognize which information about molecules of this class can be obtained by asking the right questions. The latters should be formulated in the form of particular DT structures, whose treatment is aimed at receiving a clear answer to a particular inquiry. Referring the reader to the publications mentioned above for details, we confine ourselves to a brief enumeration of the answers received.

In terms of the DT concept, all the data discussed below were obtained using a virtual device-vibrational spectrometer HF Spectrodyn [149]. The obtained IT products present one-phonon harmonic spectra of IR absorption and Raman scattering. Their deviation from empirical analogues concerns the lack of anharmonicity and a considerable blue frequency shift. The former is highly important empirically, leading to a remarkable change in the optical spectra with respect to harmonic ones [150]. Nevertheless, the main features of both IR and Raman empirical spectra of the molecular *sp*^2^ nanocarbons are of harmonic origin. Accordingly, harmonic IT product reproduces the latter well enough. As for the unavoidable blue shift of virtual harmonic frequencies [151], it is quite considerable and constitutes ~200–500 cm^−1^ in the 1000–3500 cm^−1^ region. However, it is the same for all the studied DTs and can be ignored when comparing the virtual data of the species but should be taken into account at the final stage of comparing virtual and experimental data.

Digital Twins in place of real objects, and virtual spectrometers in place of real spectrometers, are real instruments of virtual vibrational spectrometry that provides the realization of many different goals. Comparative analysis, leading to the revealing of trends, common to both virtual and experimental spectra, is the strong point of the spectrometry. The unavoidable discrepancy of obtained individual virtual spectra and experimental data is its vulnerability. Obtaining the data presented below required consideration of several tens of DTs. Their design was carried out on the original graphene domains (5,5).NGr, (9,9)Ngr, and (11,11)NGr. Digital Twins were conditionally divided into ‘monochromic’ and ‘heterochromic’ ones with respect to the chemical compositions of their necklaces. The former are carriers of highly distinguished identical chemical bonds while the latter are more adaptive to the varied chemical compositions of, say, empirical BSUs. Monochromic DTs with fully terminated edge atoms of the relevant graphene domains are of particular interest, presenting the effect of maximum contribution of the necklace heteroatoms into both the IR and Raman spectra of the molecules.

Figure 15 presents a comparative view on the spectra of monochromic DTs based on the same graphene domain. As seen in the figure, the presence of heteroatoms drastically violates the DTs’ IR spectra while leaving the Raman spectra practically unchanged. A thorough analysis of the data shown in the figure revealed that the following commonalities are inherent to all the NGMs, as well as to all of the *sp*^2^ aCs BSUs.

1. Vibrational spectra of NGMs are determined by the pool of chemical bonds, among which *sp*^2^C–C bonds configure graphene domains while *sp*^2^C–A ones (A=H, O, N, S, and so forth) shape the NGMs’ necklaces.

2. One-quantum harmonic vibration spectra of the NGMs, constituting of *N* atoms, cover 3*N*–6 vibrational modes spread over the region of 0–1800 cm^−1^, characterized for the excitation of *sp*^2^C–C bonds, and of 0–3400 cm^−1^ depending on the *sp*^2^C–A CBs configuration related to the NGM necklaces.

3. As seen in Figure 15, all the NGMs’ vibrational spectra contain a well-defined region of stretching vibrations of *sp*^2^C–C bonds in the range of 1200–1800 cm^−1^ provided with the presence of graphene domains. The stretchings are slightly disturbed in Figure 15c–e due mainly to their reaction on the oxygen atoms presence in the NGMs’ circumference (see detailed discussion of the feature in [124]).

4. The two communities of the NGMs’ chemical bonds participate in their IR and Raman spectra quite differently. Covalent homopolar *sp*^2^C–C bonds are not IR active due to a nil static dipole moment. Accordingly, IR absorption of graphene domains is extremely weak (Figure 15a) while heteropolar *sp*^2^C–A bonds show considerable absorption and are highly individually dependent on the chemical composition of the NGMs’ necklaces (Figure 15b–e). In contrast, activity of the Raman scattering is provided with *sp*^2^C–C bonds in all the cases due to which the Raman signatures of bare graphene domains as well as NGMs are similar (Figure 15a–e).

5. IR spectra, strongly dependent on the chemical composition of NGM necklaces, depend on the size of the latter rather weakly.

6. Analysis of virtual IR spectra of monochromic DTs allows for suggesting a definite set of group frequencies to be used for assignment of the spectra features to particular chemical bonds formed in the NGM necklaces.

In addition to exhibiting common trends of IR virtual spectrometry of NGMs discussed above, the first attempts were made to compare DTs’ virtual spectra with the experimental ones of the relevant *sp*^2^ aCs. Such a comparison related to TE-rGO is shown in Figure 16a. As seen in the figure, at first glance, the DRIFT spectrum of the object (red potting) differs drastically from the virtual one. However, once upshifted by 500 cm^−1^, the three-band spectrum (blue plotting) correlates well with the similar three-band virtual one, reliably supporting the atomic configuration of the DT shown in the figure. Naturally, it is impossible to speak about the complete adequacy of the proposed model to the real BSU structure, but the fact of reproduction of the main components of the latter is obvious. Thus, the evidence, provided earlier with IINS as well as with DRIFT and XPS [49] spectra, is supported with virtual spectrometry.

A completely different situation is presented in Figure 16b. In contrast to the previous case, the experimental spectrum of Ak-rGO, which deviates significantly from that for TE-rGO, is different from the virtual spectrum of DT as well. As can be seen, a similar upshift does not reveal any similarity between the real and virtual spectra, except for the band at ~1700 cm^−1^. This band is perhaps the only evidence of the presence of quinones in the structure of both real Ak-rGO BSU and DT. As for the low-frequency part of the real spectrum in the region of 500–800 cm^−1^, analysis of virtual spectra of monochromic DTs in Figure 15 as well as many others point to its attribution to hydrogen atoms in the nearest vicinity to edge atoms of the NGMs’ graphene domains. Such atoms are not presented in the DT chemical structure, which might explain the drastic discrepancy between the empirical and virtual IR spectra. As for reality, adsorbed water is the first candidate to be examined. Obviously, the presence of this water in TE-rGO may improve the fit between the empirical and virtual spectra in this region as well. The DT concept, worked out by example of NGMs [64,125], has recently been applied to the graphene oxide IR spectrum [124], positive result of which is shown in Figure 16c.

#### 5.3.3. Raman Scattering Virtual Analytics

The main features of NGMs’ Raman spectra are presented in Figure 17 in a compressed way. The Digital Twins’ analytics of Raman scattering spectra of NGMs [64,125] has revealed the following commonalities:

1. As evidenced in Figure 15 and Figure 17, the graphene domain *sp*^2^C–C stretchings determine the main pattern of the NGMs’ Raman spectra.

2. The Raman spectra of NGMs respond to the presence of heteroatoms in the molecules’ circumference due to the frequency difference of the *sp*^2^C–C and *sp*^2^C–C(A) stretchings [124] (see Figure 15). However, the violation is not critical to significantly disturb the general similarity of the Raman spectra appearance of the different *sp*^2^ aCs presented in Figure 11.

3. The Raman spectra of individual NGMs of any chemical composition are not characterized with a standard D–G doublet as seen in Figure 15.

4. The Raman spectra of individual NGMs strongly depend on their linear dimensions and are transformed from a broad-band (see Figure 17b) to narrow-band with a total domination of G-band when the size of the latter approaches the free path of graphene optical phonons $L_{ph}$ ~ 15 nm [143] (see Figure 17a). The effect, which is typical size effect of phonon spectra of molecular amorphous solids [152], strengthens when (9, 9) graphene domain is substituted with (11, 11) [64].

5. The appearance of the D-band and formation of the D–G doublet is a particular feature attributed to NGM layering (see Figure 17c). The effect becomes more pronounced when a double layer NGM is substituted with a three-layer one [64]. It is provided with the generation of dynamically stimulated *sp*^3^C–C bonds between the carbon atoms of adjacent NGM layers [124] due to the exact van der Waals contact between them. A high activity of out-of-plane stretchings of benzenoid units [118,137] makes the *sp*^3^C–C stretchings active enough to generate the D-band.

6. The D-band’s intensity increases when the number of NGM layers grows up to 4–5 nm and then markedly slows down when the stack thickness exceeds ~15 nm [142].

## 6. Express Analysis of *sp*^2^ Amorphous Carbons Based on IR and Raman Spectra

Evidently, all the above concerns the Raman spectra of real *sp*^2^ aCs, particularly, the solids’ short-range order presented with the relevant BSUs and their stacks and can be applied for analysis of empirical spectra of the solids, in general, and for their express analysis, in particular. Results of the latter, applied to a randomly selected sample [153], are demonstrated in Figure 18. Starting from the Raman spectrum in Figure 18a, one has to be convinced first in dealing with a *sp*^2^ aC, but not with graphene oxide, whose Raman spectrum looks exactly as shown in the figure [124]. Convinced by the XRPD test, we move on to read out the Raman spectrum of the rGO solid. It tells us that the solid has a stacked structure consisting of 4–5 BSUs layers, BSUs linear size being its first nanometers. A considerable width of D- and G-bands allows for suspecting a turbostratic nature of the layer packaging. Evidently, the presence of the BSUs’ necklaces is one of the reasons for both this disordering and slight increasing of the interlayer distance, as shown in Table 2. In its turn, the IR absorption spectrum in Figure 18b shows that this *rGO’s* necklace is mainly oxygenated, since none of the prominent *sp*^2^C–H models, which are concentrated below 1000 cm^−1^, are revealed. No traces of adsorbed water (region below 400 cm^−1^) are fixed as well. Following the group frequencies listed in Table 7, it is possible to suggest the oxygen-containing contributions indicated in the figure. A rather scares fine structure of the IR spectrum is in line with the attribution of marked bands below 2000 cm^−1^ to cyclic ethers, while the band at 3400 cm^−1^ evidences traces of small amount of hydroxyls in the sample. The presented express analysis of the Raman and IR absorption spectra of an rGO sample is only the beginning of the in-depth analysis and is given as an example. Nevertheless, it provides a reliable entry level of analysis requiring further confirmation using all the advanced methods of analytical chemistry and spectroscopy.

## 7. Conclusive Remarks

The first atlas of empirical and virtual analytics of *sp*^2^ amorphous carbons is over. It opens the way for further development, extension, sophistication, routing, substitution and suggests new visions and approaches. Both analytics are based on the molecular nature of the solids and are aimed at the disclosure and characterization of short-range order of the latter. The two analytics are not perfect and there are still questions to be raised and answered. What is presented for the reader’s judgment is only the first attempt to summarize the currently available ideas about *sp*^2^ amorphous carbon and offer a systematic way to check and analyze them.

## Figures and Tables

**Figure 1 nanomaterials-13-01648-f001:**
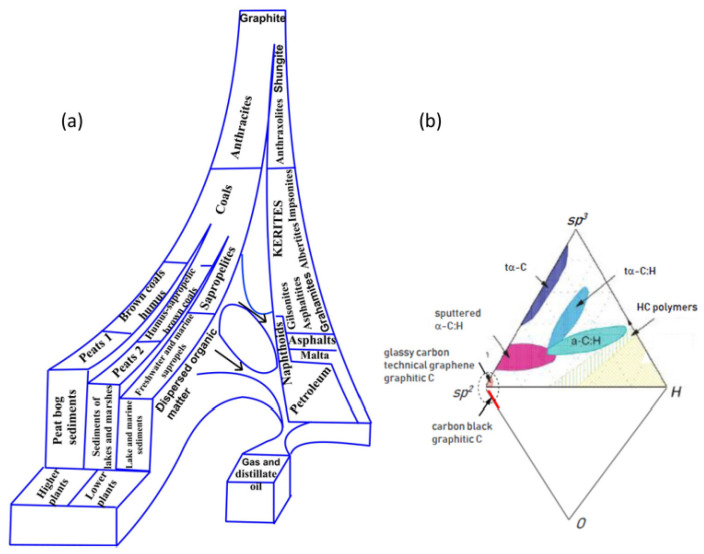
General classification of amorphous carbon. (**a**) Carbon life path in nature according to the Uspenskiy’s classification. (**b**) Rhombic diagram of synthetic amorphous carbon-hydrogen-oxygen system. Reproduced with permission from ref. [36]. Copyright 2021, Taylor & Francis Group.

**Figure 2 nanomaterials-13-01648-f002:**
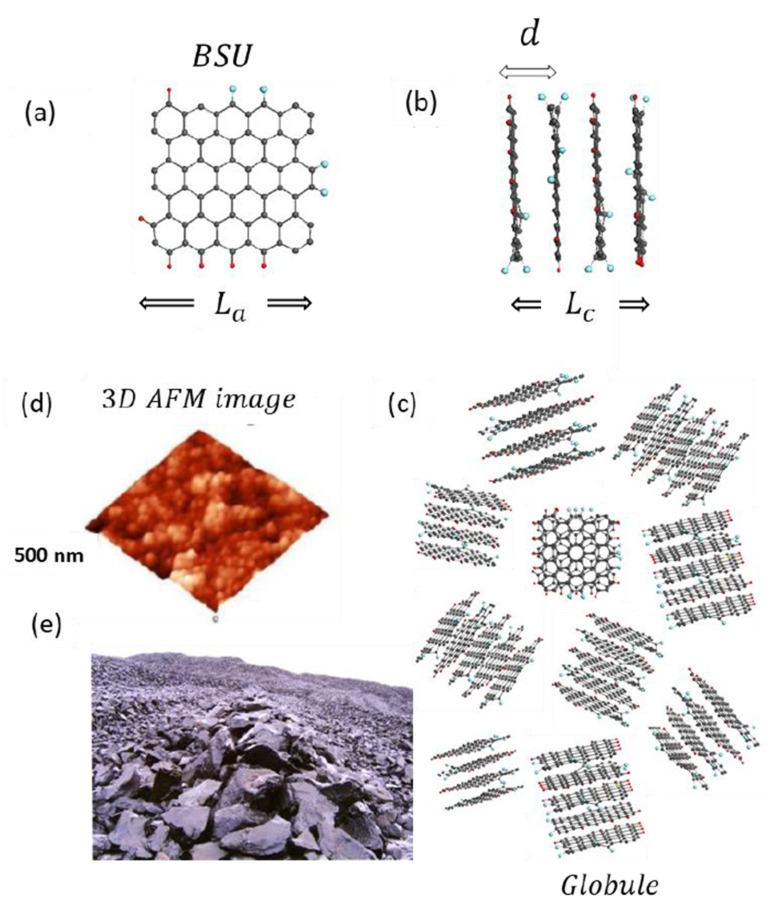
Multi-level structure of *sp*^2^ amorphous carbons, BSU size of which constitutes first nanometers. (**a**) Single necklaced-graphene BSU (C_66_O_4_H_6_ for shungite carbon [36]). (**b**) Four-layer BSU stack. *L_a_* and *L_c_* are linear size of BSU and BSU stack thickness of 1.2 nm and 1.4 nm, respectively; *d* is interlayer distance of 0.35 nm. (**c**) Planar view on a model globule composed of different stacks, the latter consisting of BSU layers from 4 to 7, differently oriented to each other, with total linear dimensions of ~6 nm [1]. Gray, red and blue balls depict carbon, hydrogen, and oxygen atoms, respectively. (**d**) 3D AFM image of globular structure of shungite carbon powder [36]. (**e**). Karelian shungite carbon deposit [63].

**Figure 3 nanomaterials-13-01648-f003:**
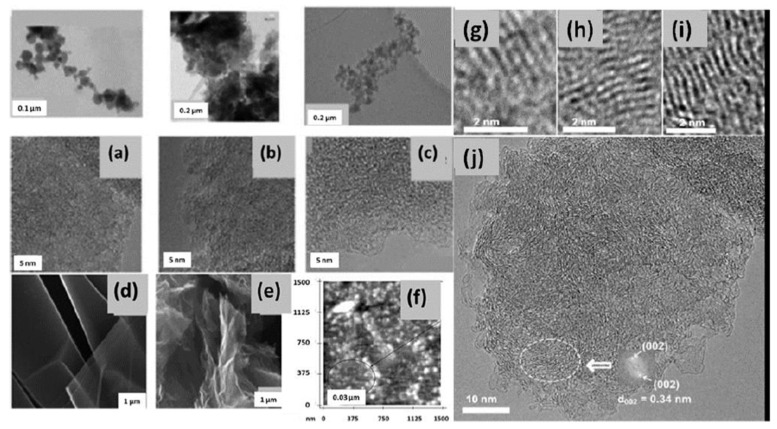
Multilevel structure of *sp*^2^ amorphous carbons in light of electron microscopy. (**a**–**c**) STEM (top) and HRTEM (bottom) images of ShC (**a**), AnthX (**b**), and CB of Sigma-Aldrich-Merk (**c**). (**d**,**e**) SEM images of paper-like technical graphenes Ak-rGO (**d**) and TE-rGO (**e**). Adapted from Ref. [49]. (**f**) SEM image of globules’ aggregates [7]. (**g**–**j**) HRTEM images of the atomic structure of ShC: stacks of flat (**g**,**h**) and bent (**i**) BSU layers. General view of a ShC particle (**j**), for which Fourier diffraction pattern of the indicated area (see inset) was obtained. Adapted from Ref. [13].

**Figure 4 nanomaterials-13-01648-f004:**
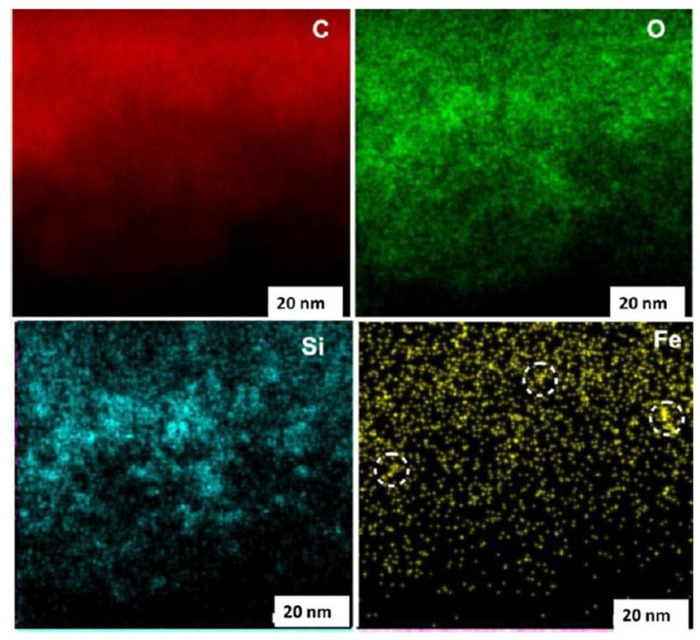
HRTEM element mapping of *sp*^2^ amorphous carbons. Shungite carbon in the light of C, O, Si, and Fe elements. The circles in the iron map indicate nanoclusters.

**Figure 5 nanomaterials-13-01648-f005:**
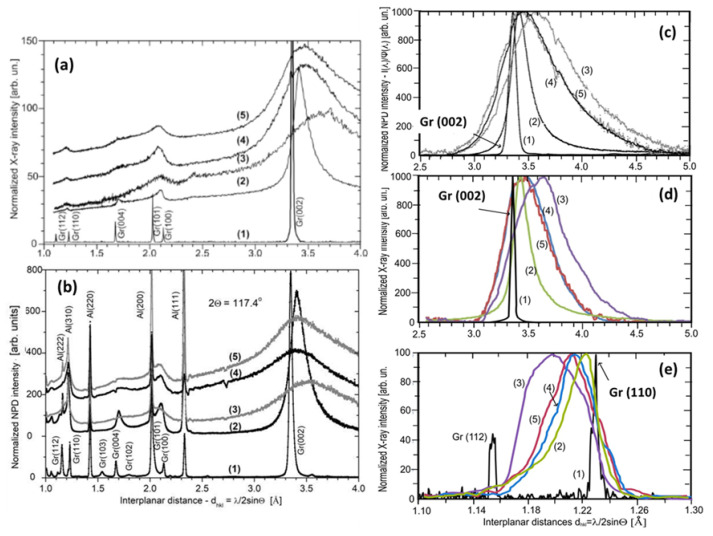
Powder diffraction in the service of short-range structure determination of *sp*^2^ amorphous carbons. (**a**) Panoramic views of XRPD plottings of *sp*^2^ aCs and graphite. (**b**) The same related to NPD. (**c**) Normalized intensities of the NPD Gr (*002*) reflexes. (**d**) The same concerning XRPD. (**e**) Normalized intensities of the XPRD Gr (*110*) reflexes. (1)–(5) mark Gr, CB624, CB632, ShC, and AntX. See the sample nomination in Table 1 and plottings details in Ref. [48]. Graphite of Botogol’sk deposit [81] was used as standard.

**Figure 6 nanomaterials-13-01648-f006:**
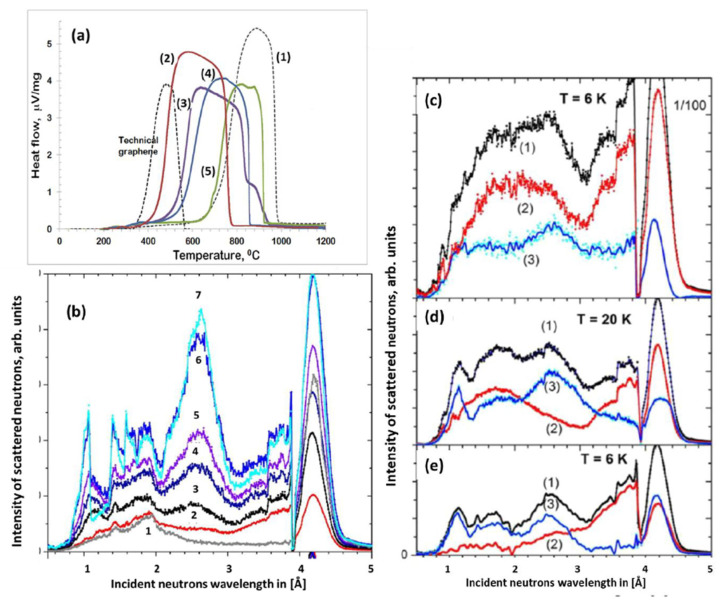
Hydrogen detection in *sp*^2^ amorphous carbons. (**a**) DSC thermograms of as prepared solids: spectral Gr (1), ShC (2), AntX (3), CB632 (4), and CB624 (5) [48]. DSC curve of the lab rGO is reconstructed from experimental data presented in Ref. [88]. (**b**) Time-of-flight IINS spectra of water gradually added to CB632. 1, 2, 3 mark scattering from Al-cryostat, dry and as prepared samples, respectively. (4–6) Spectra of water loaded in 100 g of as prepared CB632 sample in mL: 0.5 (4), 1.0 (5) and 2.0 (6). (7) 2.5 ml of water loaded in as prepared CB624 sample. (**c**–**e**) Normalized TOF IINS spectra of *sp*^2^amorphous carbons after subtracting the Al-cryostat background: AntX (**c**), ShC (**d**), CB632 (**e**). Digits mark spectra of as prepared (1) and dry (2) samples as well as adsorbed water (3), the latter obtained as the difference between spectra (1) and (2) (see details in [47]).

**Figure 7 nanomaterials-13-01648-f007:**
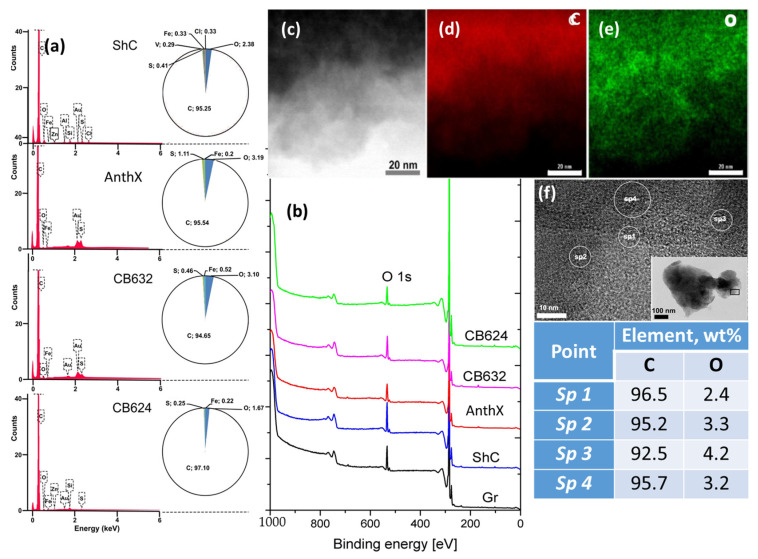
Oxygen detection in *sp*^2^ amorphous carbons. (**a**) EDS spectra and content diagrams of dry *sp*^2^ amorphous carbons. Spherical diagrams visualize the samples chemical content, details of which are given in Ref. [48]. (**b**) XPS survey spectra of as prepared amorphous carbons and graphite GSM2 at room temperature [48]. (**c**,**d**) Dark-field HRTEM of an edge fragment of ShC particle (**c**) and its element EDS mapping for C (**d**) and O (**e**) elements. (**f**) Localization of EDS point elemental analysis of ShC, the data of which are listed in table below. The inset marks the region under study. Details are given in [13].

**Figure 8 nanomaterials-13-01648-f008:**
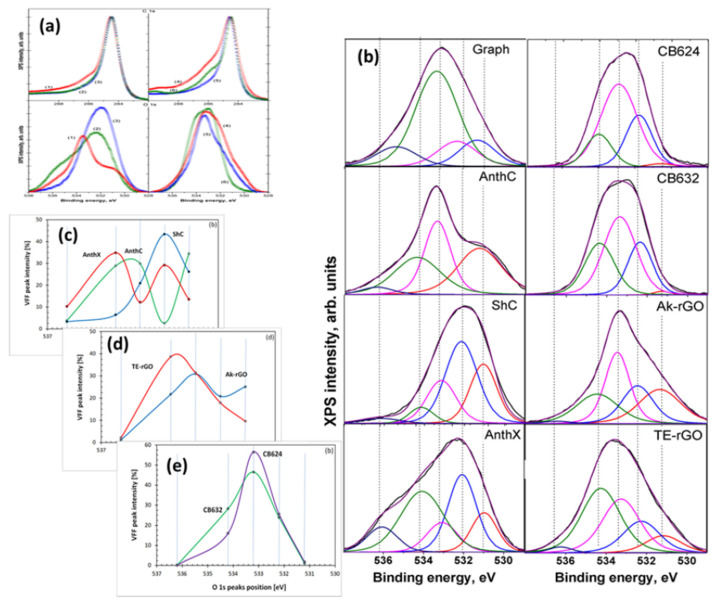
*sp*^2^ Amorphous carbon chemical bonds configuration in view of XPS. (**a**) C1s (top) and O1s (bottom) XPS spectra of AnthC (1); AnthX (2); ShC (3); TE-rGO (4), Ak-rGO (5) and CB632 (6) at room temperature. (**b**) Expanded O1s XPS spectra and fraction-distributions of Voigt-fitting-function (VFF) peaks of O1s spectra over peaks number of natural (left) and synthetic (right) *sp*^2^ amorphous carbons at room temperature. (**c**–**e**) Distribution of the VFF peaks intensity over group binding energies for natural (**c**), synthetic (**d**) and industrial (**e**) *sp*^2^ aCs. Details are presented in [49].

**Figure 9 nanomaterials-13-01648-f009:**
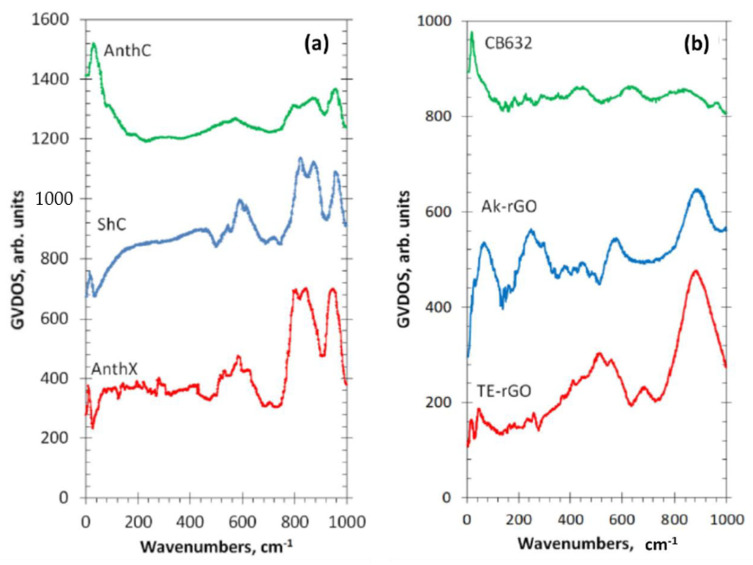
Generalized density of state of the hydrogen-involved vibrations of *sp*^2^ amorphous carbons. (**a**) GVDOS spectra of dry natural solids, derived from IINS spectra at T = 20 K. (**b**) The same but for industrial and synthetic solids. See details in Ref. [49]. Reproduced with permission from ref. [49]. Copyright 2020, Taylor & Francis Group.

**Figure 10 nanomaterials-13-01648-f010:**
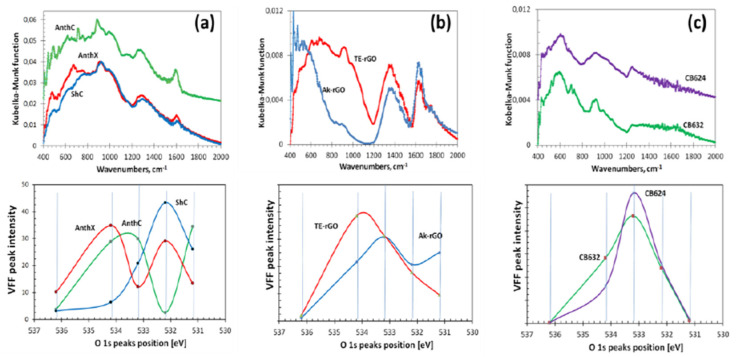
Joint DRIFT-XPS spectral analysis of *sp*^2^ amorphous carbons. Top. DRIFT spectra of amorphous carbons at room temperature: natural (**a**), synthetic (**b**), and industrial (**c**) solids. Bottom. Distribution of intensities of VFF peaks of the O1s spectra of the solids presented in Figure 8b over group binding energies listed in Table 4. Adapted from [49].

**Figure 11 nanomaterials-13-01648-f011:**
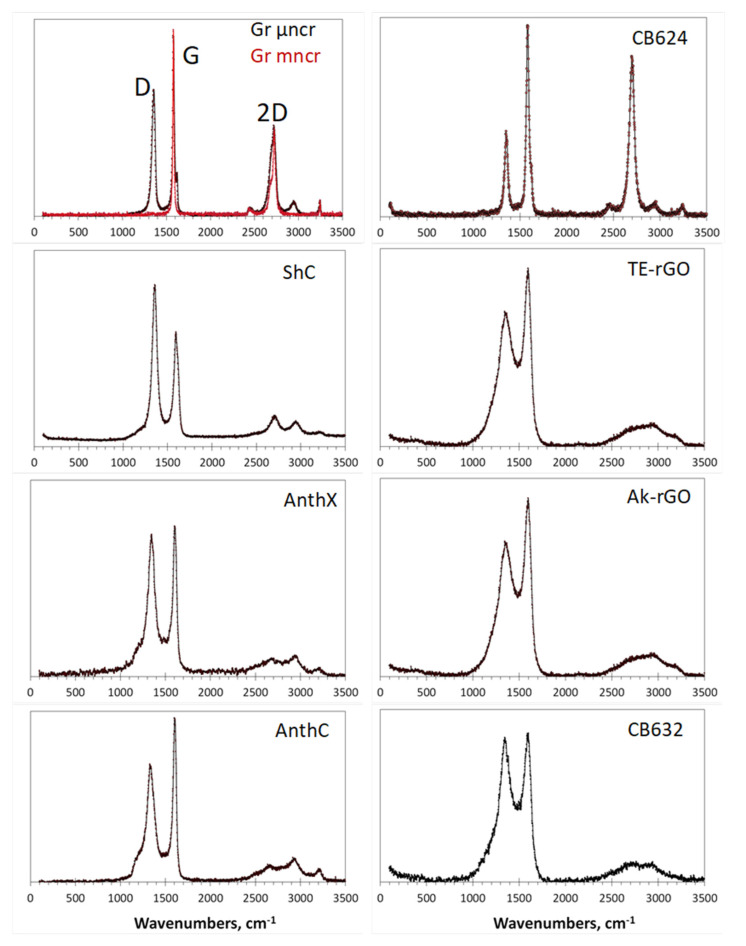
Raman spectra of *sp*^2^ amorphous carbons at room temperature: shungite carbon (ShC), anthraxolite (AnthX), anthracite (AnthC), technical graphene TE-rGO and Ak-rGO, carbon blacks CB632 and CB624, as well as mono- (mncr) and micronanocstructured (μncr) Botogol’sk graphites, respectively (see details in [50]).

**Figure 12 nanomaterials-13-01648-f012:**
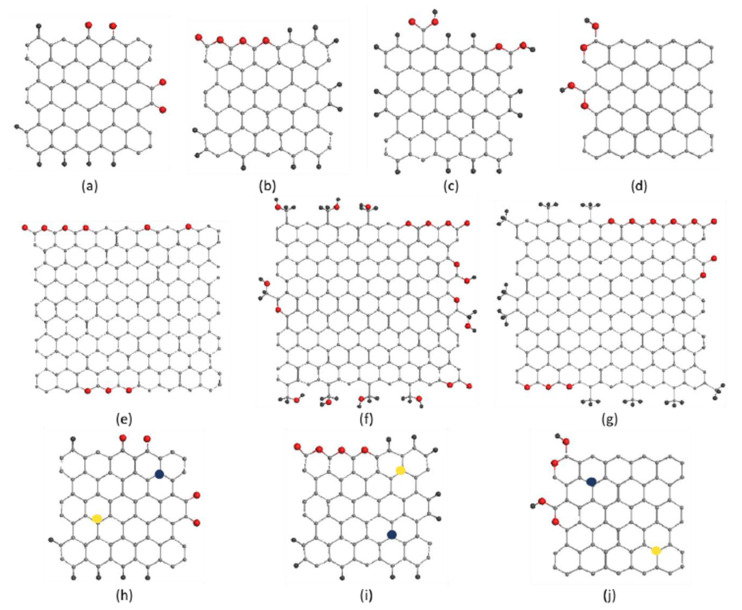
Molecular models of basic structure units of *sp*^2^ amorphous carbons. Equilibrium structures of {*CHO*}-triad species: ShC (**a**), AnthX (**b**), AnthC (**c**), CB632 (**d**), CB624 (**e**), TE-rGO (**f**) and Ak-rGO (**g**). {CHO}-triad species complemented with nitrogen and sulfur additives: ShC (**h**), AnthX (**i**) and CB632 (**j**). Atomic content of the models follows the data listed in Table 3. Gray, red, black (**a**–**g**), completed with dark blue, yellow (**h**–**j**) balls depict carbon, hydrogen, oxygen, nitrogen and sulfur atoms, respectively.

**Figure 13 nanomaterials-13-01648-f013:**
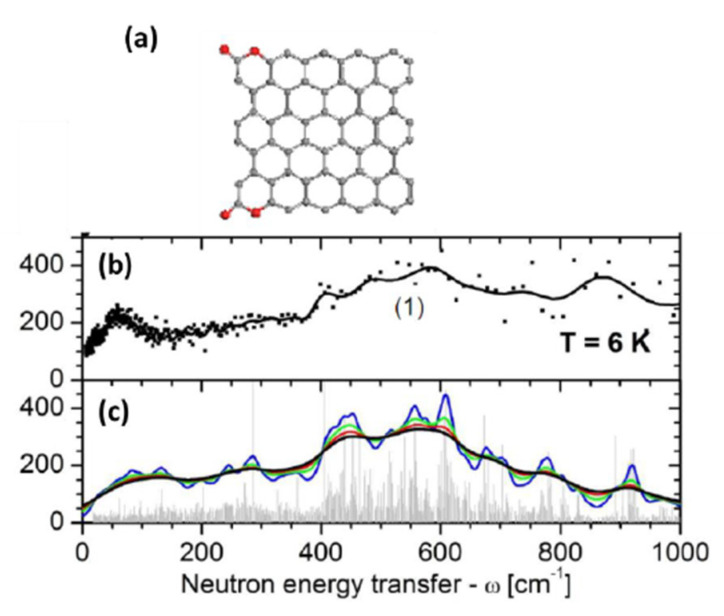
Adsorbed water in *sp*^2^ amorphous carbon. (**a**) NGM model of BSUs of carbon blacks. (**b**) IINS-spectra-derived GVDOS of adsorbed water in as prepared CB632. (**c**) Virtual GVDOS of four-hydrogen-bond configured molecules of retained water. Gray and red balls depict carbon, and oxygen, respectively.

**Figure 14 nanomaterials-13-01648-f014:**
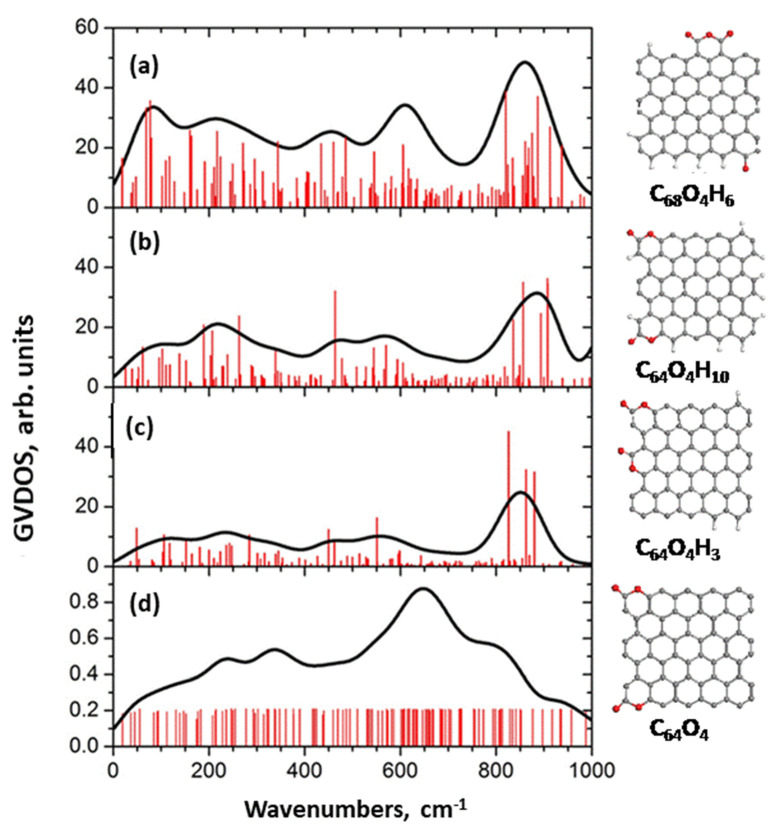
Virtual GVDOS spectra of graphene oxyhydrides related to the following BSUs’ models: (**a**) ShC—C_68_O_4_H_6_, (**b**) AnthX—C_64_O_4_H_10_, (**c**) CB632—C_64_O_4_H_3_, and (**d**) CB624—C_64_O_4_. Original bars are convoluted by Gaussian of 80 cm^−1^ half-width. Gray, white and red balls depict carbon, hydrogen and oxygen atoms, respectively.

**Figure 15 nanomaterials-13-01648-f015:**
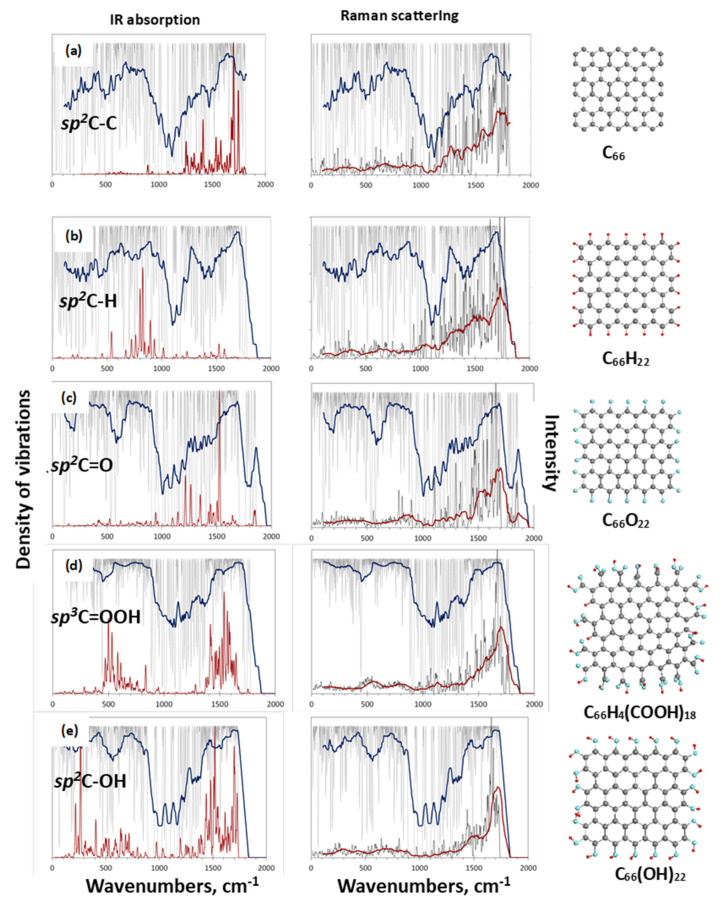
Virtual vibrational spectra (light gray) as well as one-phonon IR absorption and Raman scattering spectra (gray) of a set of the (5,5)NGr-based monochromic digital twins with fully terminated edge atoms of the graphene domains (**a**–**e**) [64]. Original sticks of the spectra are convoluted with Gaussian bandwidth of 10 cm^−1^. Plottings of densities of vibrations and Raman spectra are accompanied with trend lines corresponding to 50–point linear filtration, dark blue and red, respectively. HF Spectrodyn virtual spectrometer. Equilibrium DT structures supplied with chemical formulas are presented on the right.

**Figure 16 nanomaterials-13-01648-f016:**
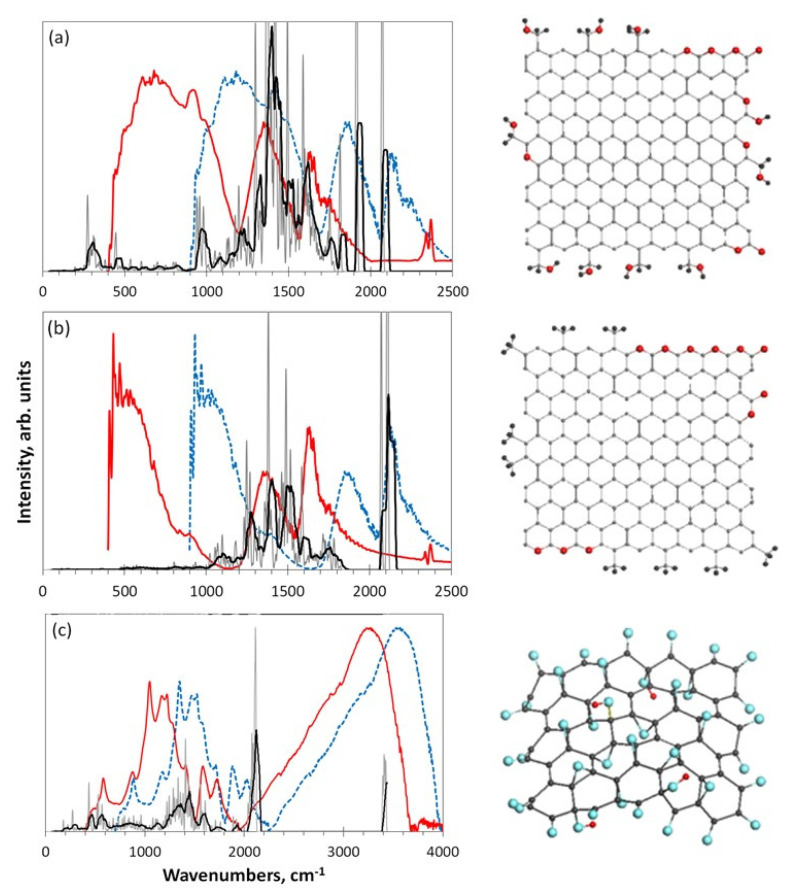
Digital twins’ analytics of IR spectra of reduced and parental graphene oxides. (**a,b**) Virtual one-phonon (gray, supplemented with black trend lines corresponding to 50–point linear filtration, HF Spectrodyn) as well as original (red) and 500 cm^−1^ blue-shifted (blue) experimental IR absorption spectra of TE-rGO and Ak-rGO, respectively [64]. (**c**) The same but original and 300 cm^−1^ blue-shifted experimental IR spectra of graphene oxide [124]. Equilibrium structures of rGO’s DTs C_185_H_28_O_19_ and C_181_H_27_O_11_ (see Table 8) as well as of GO’s DT C_66_O_40_H_4_ are shown on the right.

**Figure 17 nanomaterials-13-01648-f017:**
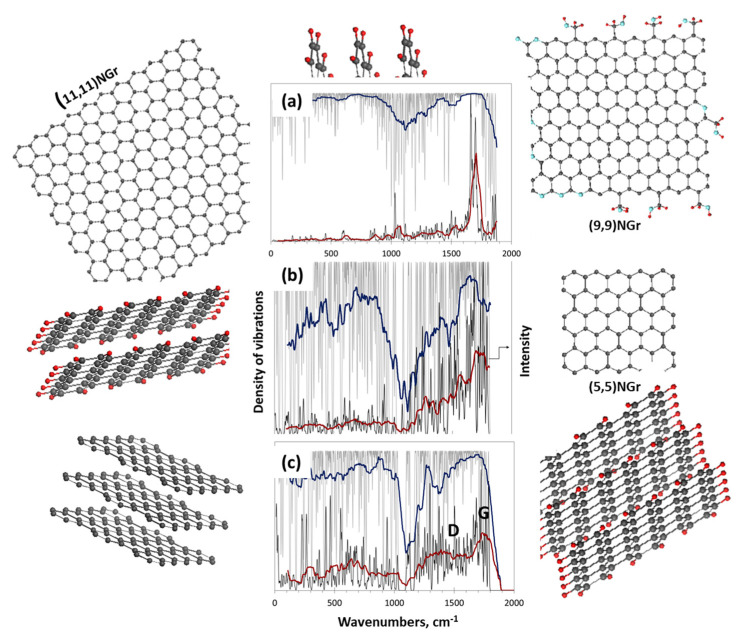
Digital twins’ analytics of Raman spectra of individual and layered NGMs of different size and compositions. (**a**,**b**). Virtual densities of vibrations (light gray) and one-phonon Raman spectra (gray) of one-layer TE-rGO (C_192_O_19_H_44_) and bare domain (5,5) NGr (C_66_), respectively. (**c**) The same but two layers of hydrogenated domain (5, 5) NGr (C_132_H_44_). Original sticks of the spectra are convoluted with Gaussian bandwidth of 10 cm^−1^. Both densities of vibrations and Raman spectra are accompanied with 50-points trend lines, dark blue and red, respectively. HF Spectrodyn. Adapted from Refs. [64,125].

**Figure 18 nanomaterials-13-01648-f018:**
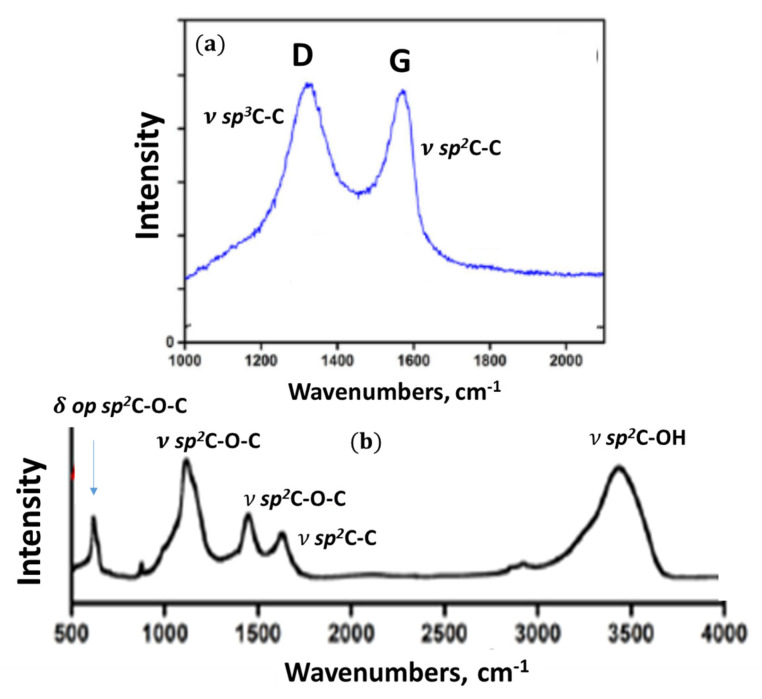
Express analysis of *sp*^2^ amorphous carbon. Raman scattering (**a**) and IR absorption (**b**) spectra of a lab-fabricated reduced graphene oxide [153] in notations of vibrational analytics (see text).

**Table 1 nanomaterials-13-01648-t001:** *sp*^2^ Amorphous carbons selected for a joint study.

No	Samples	Abbreviation	Origin	References
1	Shungite carbon	ShC	Shun’ga deposit of Karelia, Russia	[7]
2	Anthraxolite	AnthX	Pavlovsk deposit of Novaya Zemlya, Russia	[8]
3	Anthracite	AnthC	Donetsk deposit, Russia	[9]
4	rGo	Ak–rGO	Institute of the Inorganic Chemistry, RAS, Moscow, Russia	[10]
5	rGO	TE–rGO	Institute of the Chemical Physics, RAS, Moscow, Russia	[11]
6	Black carbon	CB632	Sigma-Aldririch-Merk company, USA	[12]
7	Black carbon	CB624	Sigma-Aldririch-Merk company, USA	[12]

**Table 2 nanomaterials-13-01648-t002:** Parameters of the short-range structure of *sp*^2^ amorphous carbons ^1^.

Samples	d (Å)	$L_{c}$, nm	Number of BSU Layers	$L_{a}$, nm	Ref
ShC	3.47(n); 3.48(X)	2.5(n); 2.0(X)	7(n); 5–6(X)	2.1(X)	[48]
AnthX	3.47(n); 3.47(X)	2.5(n); 1.9(X)	7(n); 5–6(X)	1.6(X)	[48]
AnthC (Donetsk)	3.50(X)	2.2(X)	5–6(X)	2.1(X)	[49]
Ak-rGO	3.50(n)	2.4	7(n)	>20	[5]
TE-rGO	3.36(n)	2.9	8(n)	>20	[12]
CB632	3.57(n); 3.58(X)	2.2(n); 1.6(X)	6(n); 4–5(X)	1.4(X)	[48]
CB624	3.40(n); 3.45(X)	7.8(n); 4.1(X)	23(n); 12(X)	2.5(X)	[48]
μncr Gr	3.35	>20^2^	~100	>20	[48]

^1^ Notations (n) and (X) indicate NPD and XRPD data, respectively; ^2^ The definition “> 20 nm” marks the low limit of the dimension pointing that it is bigger than the CSR of crystalline graphite equal to ~20 nm along both *a* and c directions. Actual dimensions are of micrometer range.

**Table 3 nanomaterials-13-01648-t003:** Chemical content of BSUs of *sp*^2^ amorphous carbons.

Samples	Elemental Analysis, wt%						XPS Analysis, wt%			
	C	H	N	O	S	Ref.	C	O	Minor Impurities	Ref.
ShC	94.44	0.63	0.88	4.28	1.11	[48]	88.5	8.6	2.9	[48]
AnthX	94.01	1.11	0.86	2.66	1.36	[48]	89.5	7.7	2.8	[48]
AnthC	90.53	1.43	0.74	6.44	0.89	[49]	89.6	8.1	2.3	[49]
TE-rGO	84.51	1.0	0.01	13.5	1.0	[49]	82.3	14.8	2.9	[49]
A_K_-rGO	89.67	0.96	0.01	8.98	0.39	[49]	89.5	7.6	2.9	[49]
CB624	99.67	0.18	0	0.15	-	[48]	93.1	5.9	1.0	[48]
CB632	97.94	0.32	0.04	1.66	0.68	[48]	90.7	7.8	1.5	[48]

**Table 4 nanomaterials-13-01648-t004:** Group binding energies of O1s XPS spectra attributed to BSUs of *sp*^2^ amorphous carbons (composed from the published data [113]).

GBEs	BE, eV	Assignments
1	531.2	C=**O**, **O**=C–O–C=**O**, **O**=C–O–C (lactones and pairs of lactones)
2	532.2	O=C–**O**–C (lactones); **O**=C–C=**O** (o-quinones); **O**=C–OH; C=**O** in aggregated cyclic ethers with lactone
3	533.2	*sp*^2^C–**O**H; C–**O**–C in cyclic ethers; C–O–C–**O**H (hydroxypyrans: singles and pairs); O=C–**O**–C (lactones and pairs of lactones); O=C–**O**H; C–**O**–C in aggregated cyclic ethers with lactones
4	534.2	C–**O**–C in aggregated cyclic ethers; C–**O**–C–OH (hydroxypyran: singles and pairs); C–O–C in aggregated cyclic ethers with lactones
5	536.2	O=C–O–C–**O**–C–**O**–C–O–C=O in aggregated cyclic ethers with lactones

**Table 5 nanomaterials-13-01648-t005:** XPS-revealed oxygen-containing groups related to BSUs of *sp*^2^ aCs, based on data of Refs. [48,49].

**ShC**	carbonyls *sp*^2^C=O; acid anhydride O=C–O–C=O; *o*-quinone O=sp^2^C–sp^2^C=O, carboxyls *sp*^2^C=OOH.
**AntX**	hydroxyls *sp*^2^–OH; C–O–C–OH (hydroxypyran-HP) and pairs of HPs; C=OOC(lactone) and pairs of lactones; aggregated cyclic ethers with lactones.
**AntC**	carboxyls *sp*^2^C–COOH; cyclic ethers; aggregated cyclic ethers; pyran and hydroxypyran.
**Ak-rGO**	aggregated cyclic ethers and aggregated cyclic ethers with lactones; lactones and pairs of lactones.
**TR-rGO**	aggregated cyclic ethers and aggregated cyclic ethers with lactones; hydroxypyrans and lactones, both singles and pairs.
**CB632**	C–O–C in cyclic ether and aggregated cyclic ether; C–O–C of pairs of cyclic ether and aggregated cyclic ether with lactone.
**CB624**	C–O–C in cyclic ether, aggregated cyclic ether and aggregated cyclic ether with lactone

**Table 8 nanomaterials-13-01648-t008:** Chemical formulas of analytically provided models of basic structural units of *sp*^2^ amorphous carbons.

No	Samples	Chemical Formula
1.	ShC	C_66_H_6_O_4_
2.	AnthX	C_66_H_10_O_4_
3.	AnthC	C_66_H_14_O_4_
4.	Ak-rGO	C_181_H_27_O_11_
5.	TE-rGO	C_185_H_28_O_19_
6.	CB632	C_66_H_2_O_4_
7.	CB624	C_181_O_9_

## Data Availability

Any data or material that support the findings of this study can be made available by the corresponding author upon request.

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
