# Peer review of "A Neoteric View of sp2 Amorphous Carbon"

_nanomaterials, 2023, doi:10.3390/nano13101648_

Round 1

Reviewer 1 Report

The major problem with this work is the ENGLISH LANGUAGE.

The author must ask an English native for proofreading. Not only the lalnguage, but athe sentences are not scientifically composed. Unusual word are used.

A few examples:

1.: Main title: Neoteric -> Newer? Not good word as well as not explained in the Abstract...

2.: Chapter title: concise historical introduction  -> introduction

3.: Introduction:

  1st sentence: Presented is a concentrated synopsis of facilities of empirical and virtual analytics that,

once applied, have provided a fully new vision of sp2 amorphous carbons. 

  --->  Empirical and virtual analytics is presented providing a new view of sp2 amorphous carbons. 

4.: Intro chapter 1st sentence : 

  The last two decades have seen a profound breakthrough in our understanding of sp2 23 amorphous carbon (aC). 

  ---> Profound breakthrough have been seen in the last two decades in our understanding of sp2 amorphous carbon (aC). 

IN THE PRESENT FORM IT IS NOT READABLE, TO REVIEW IT MUST BE REWRITTEN.

Reviewer 2 Report

While this could be an interesting work of Sheka, the flow of writing is not very smooth. There are many grammatical errors scattered all around the ms. Therefore, I urge the author to check the ms with a native English writer. My other comments follow:

1-      Picture quality (viz. Fig. 1, 7, 11, 16, 18 and 5) is too poor. It is very difficult to read the text shown in (a) and (b). It should be improved. Clearly, all these figures should be significantly improved during a revision since they are unacceptable in the current form.

2-      Some figures taken from other studies are not even cited correctly (see Fig. 4, for example).  

3-      The conclusion part of the long review is too week. There is no future direction for readers that they can attempt cultivate to the next step. Moreover, the author should explain with transparency “what is the modern view” provided for sp2 carbon?  

4-      Although there is no discussion of the underlying chemistry of infrared spectroscopy, the ms contains a large portion of discussion involving IR spectral features.

5-      Many terms introduced in the ms without clarifying them. For example, the term such as “IINS …” can be sharply unclear to the readers of journal.

6-      The paper is too long, thus should be reduced to a maximum of 20-25 pages long. Otherwise, it is very tiring!!

7-      The data in Table 1 show the involvement of researchers from Russia and USA. What about the rest of the world?

8-      The objective the review is not transparent. And the word “Neoteric view” in the title of the ms should be replaced by terms such as “modern view” to make it simpler for readers.

Reviewer 3 Report

Dear editor,

the review is mainly focused on the published works of the author, especially the second part about modeling and digital Twins. I am not able to devise a suitable opinion.

Round 2

Reviewer 1 Report

nanomaterials-2160496, review on: E. F. Sheka: A Neoteric View of sp2 Amorphous Carbon

  The authors should emphasize in the Abstract and the introduction part, that it is a review article, and must emphasize that which of the measurements 

in many figures and tables are their result or collected from literature.  

  I still do not like the word "Neoteric" in the title... 

  Abbreviation "basic structural unit (BSU)" must be written in the Abstract too, because difficult to understand otherwise. 

  As I mentioned, the article needs serious langueage editing, e.g.:

     line 30: "In the rays of these two illuminations," -> In view of these, 

     line 33: "My first personal encounter with sp2 aC was in 2005..." <- This is not a novel, but a scientific articel.

     line 40: fairy tale-like stories...

     line 59.: Chemistry teaches us that covalent bonds...

     line 74.: However, one day, on the desk of one of my colleagues, I saw two flasks filled with black powder

     etc., etc., 

  The amrphous carbon (aC) shoud not be used as abbreviation! It should be written out... 

  Small list of abreviations should be written in the end of the article in Reference section. 

  line 22-101: A concise historical introduction -> A historical introduction,

               more importantly, it shold be reduced to half page, there are full of trivial information.

  ref. 63.: the long number at the end of line is unnecessary!, somethong wrong with that citation!

  reference 65: Wrong journal name: correctly: J of Carbon Reserach C

  line 950, ref. 149: It is fundamental to cite the publication in relation to this part of the work:

    Electronic Schrodinger Equation Directly for Ground State One-Electron Density and Electronic Energy, 

       International Journal of Quantum Chemistry 113 (2013) 1479-1492, 

  line 1002, ref. 151: For virtual harmonic frequencies related to amorphous carbons, the authora must cite: 

    zero point energy on the nuclear frame and number of electrons in molecular systems, 

      Journal of Molecular Structure: THEOCHEM, 712 (2004) 153-158, 

Reviewer 2 Report

The author of the study seems to be uncomfortable with my comments. He/she did not address my questions in a well-defined manner. Most of my concerns were ether overlooked, or answered carelessly. I did not see any seriousness of the author to improve the quality of the ms. This can readily be seen from the text copied below:

Reviewer's concern: The conclusion part of the long review is too week. There is no future direction for readers that they can attempt cultivate to the next step. Moreover, the author should explain with transparency “what is the modern view”, as provided for sp2 carbon?

Author's reply: The conclusion emphasizes the atlas-like format of the review and is quite full.

The same style is applicable to my other concerns, and hence there is no revisions made to the ms.

Accordingly, I treat this work of substandard quality, and hence I do not recommend publication of this work in its current form
